# Mammalian SWI/SNF chromatin remodeler is essential for reductional meiosis in males

Debashish U. Menon [1], Oleksandr Kirsanov[2], Christopher B. Geyer[2,3] & Terry Magnuson [1✉]

The mammalian SWI/SNF nucleosome remodeler is essential for spermatogenesis. Here, we identify a role for ARID2, a PBAF (Polybromo - Brg1 Associated Factor)-specific subunit, in meiotic division. *Arid2cKO* spermatocytes arrest at metaphase-I and are deficient in spindle assembly, kinetochore-associated Polo-like kinase1 (PLK1), and centromeric targeting of Histone H3 threonine3 phosphorylation (H3T3P) and Histone H2A threonine120 phosphorylation (H2AT120P). By determining ARID2 and BRG1 genomic associations, we show that PBAF localizes to centromeres and promoters of genes known to govern spindle assembly and nuclear division in spermatocytes. Consistent with gene ontology of target genes, we also identify a role for ARID2 in centrosome stability. Additionally, misexpression of genes such as *Aurkc* and *Ppp1cc* (*Pp1γ*), known to govern chromosome segregation, potentially compromises the function of the chromosome passenger complex (CPC) and deposition of H3T3P, respectively. Our data support a model where-in PBAF activates genes essential for meiotic cell division.

[1] Department of Genetics, and Lineberger Comprehensive Cancer Center, The University of North Carolina at Chapel Hill, Chapel Hill, NC 27599-7264, USA. [2] Department of Anatomy & Cell Biology at the Brody School of Medicine, East Carolina University, Greenville, NC 27858, USA. [3] East Carolina Diabetes and Obesity Institute, East Carolina University, Greenville, NC 27858, USA. ✉email: trm4@med.unc.edu

Generation of haploid gametes involves meiotic recombination during prophase-I followed by two successive rounds of cell division (MI and MII). Several known chromatin regulators have been implicated in male meiosis[1–4]. We previously identified a requirement for BRG1, the catalytic subunit of mammalian SWI/SNF chromatin remodeling complex, during spermatogenesis[5]. Briefly, SWI/SNF ensures meiotic progression by activating key spermatogonial and meiotic genes while repressing somatic genes in the germline[6].

In addition to BRG1, multiple subunits (~10–14) constitute SWI/SNF, potentially giving rise to biochemically heterogenous subcomplexes[7]. So far, studies have identified at least three distinct subcomplexes, namely, BAF (*Brahma/Brg1* Associated Factor), PBAF (*Polybromo*-BAF), and ncBAF (noncanonical BAF)[8]. The identities of these complexes are based on the presence of mutually exclusive subunits. Notably, the ARID (AT-rich interaction domain) containing putative DNA binding subunits[9] ARID1A/1B and ARID2 identify BAF and PBAF respectively, while ncBAF is associated with GLTSCR1/1 L (Glioma tumor suppressor candidate region gene 1/1-Like) subunits[8]. SWI/SNF subcomplexes with specialized subunit compositions govern distinct tissue or cell-specific developmental programs such as embryonic stem (ES) cell pluripotency and differentiation as well as neuronal and cardiac cell fate specification[10–12]. Whether similar mechanisms dictate SWI/SNF functions during gametogenesis remains unknown.

In this study, we show that ARID2, a PBAF-specific subunit, is essential for reductional meiosis, a process by which parental genomes are halved at the end of MI. ARID2 facilitates metaphase-I exit by promoting spindle assembly and chromosome segregation. ARID2 influences centrosome formation, the kinetochore association of the known regulator of anaphase onset Polo-like kinase1 (PLK1), the centromeric targeting of Histone H3 threonine3 phosphorylation (H3T3P)/histone H2A threonine120 phosphorylation (H2AT120P), and localization of the chromosome passenger complex (CPC). The expression of key genes associated with these processes are directly or indirectly regulated by ARID2. Our studies reveal a mechanism where-in PBAF-directed gene regulation underlies its role in meiotic cell division.

## Results

**ARID2 is essential for reductional meiosis.** To explore whether SWI/SNF subcomplexes might selectively govern distinct germ cell processes, we first examined existing RNA-seq data generated from purified spermatogenic cell populations[13] and monitored the temporal expression profiles of SWI/SNF subunits that define two common subcomplexes, namely BAF and PBAF (Fig. 1a). Of particular interest were the ARID-containing SWI/SNF subunits known to bind DNA with loose specificity[9]. Among the subunits examined, the mRNA abundance of the PBAF DNA binding subunit[14] *Arid2* drew our attention as it peaked at pachynema (Fig. 1a). More recent single-cell RNA-seq data identified *Arid2* as an early pachytene marker[15], suggesting a role in meiosis-I. We, therefore, performed immunostaining on testis cryosections to profile ARID2 abundance in prophase-I spermatocytes staged by γH2Ax. Consistent with its mRNA profile, ARID2 protein levels increased from undetectable to elevated during the transition from early prophase-I [preleptonema (PL)/zygonema (Z)] to late prophase-I [pachynema (P)/diplonema (D)] (Fig. 1b). Additionally, ARID2 was also enriched proximally to chromocenters in haploid round spermatids (Fig. 1b). To investigate its role in meiosis, we conditionally deleted *Arid2* using the spermatogonia-specific *Stra8-Cre* transgene[16], which results in significant depletion of ARID2 and smaller testes in *Arid2^{cKO}* relative to

*Arid2^{WT}* (Fig. 1c and Supplementary Fig. 1a). A histological examination revealed a striking loss of haploid spermatid cell populations and absence of sperm in *Arid2^{cKO}* relative to *Arid2^{WT}* adult seminiferous tubule lumina and cauda epididymides, respectively (Fig. 1d and Supplementary Fig. 1b). The *Arid2^{cKO}* testes were characterized by an accumulation of stage XII tubules (85%), indicative of a meiotic arrest (Fig. 1d). While meiotic prophase-I appeared unhindered, we noticed a striking accumulation of metaphase-I spermatocytes in the *Arid2^{cKO}* testis (Fig. 1d, e and Supplementary Fig. 1c). The near absence of secondary spermatocytes and spermatids in the *Arid2^{cKO}* testis indicates an essential role for ARID2 in meiotic cell division.

**ARID2 influences spindle assembly and chromosome organization.** The inability of *Arid2^{cKO}* spermatocytes to progress beyond metaphase-I prompted us to investigate the influence of ARID2 on critical processes regulating chromosome segregation. These include chromosome condensation, alignment, and microtubule–kinetochore attachment.

We began by monitoring, in *Arid2^{WT}* and *Arid2^{cKO}* seminiferous tubules, a known marker of chromosome condensation, phosphorylated histone H3 on serine 10 (H3S10P), as well as the spindle marked by β-Tubulin. While H3S10P levels between *Arid2^{WT}* and *Arid2^{cKO}* metaphase-I spermatocytes appeared similar, certain mutant metaphase-I chromosomes were rounded and failed to congress at the metaphase plate (Fig. 2a, row 3). These defects prompted us to examine meiotic metaphase spreads where we observed no signs of pairing abnormalities or aneuploidy in *Arid2^{cKO}* spermatocytes (Supplementary Fig. 2a). More strikingly, 52% of the scored metaphase-I spermatocytes from *Arid2^{cKO}* tubules displayed a deficiency in spindle fibers (Fig. 2a, row 3). Metaphase-I spermatocytes that lacked a spindle were demonstrably deficient in ARID2 relative to those that displayed a spindle in *Arid2^{cKO}* tubules (Supplementary Fig. 2b). The latter therefore represent cells that underwent inefficient Cre-mediated deletion and were, therefore, considered internal controls. In sum, *Arid2^{cKO}* metaphase-I spermatocytes feature aberrant spindle assembly and abnormal chromosome organization.

**ARID2 regulates PLK1 associations at kinetochores.** Normally, a failure to establish microtubule–kinetochore attachments would activate the spindle assembly checkpoint (SAC), which delays metaphase exit[17,18]. Since *Arid2^{cKO}* spermatocytes are deficient in microtubules we were curious to determine the status of SAC proteins, many of which associate with kinetochores. For this purpose, we examined Polo-like kinase1 (PLK1), a well-known SAC regulator[19,20] that also influences spindle assembly[21–26]. In control (*Arid2^{Het}*) metaphase-I spermatocytes, PLK1 foci overlapped with Centromere protein A (CENPA), a known histone H3 variant and component of functional centromeres[27] (Fig. 2b). In contrast, 60% of the scored metaphase-I spermatocytes in *Arid2^{cKO}* tubules lacked centromeric PLK1 (PLK1^−, Fig. 2b). Those that still displayed centromeric PLK1 (PLK1^+) appeared to express ARID2 at almost normal levels, indicating that they were internal controls (Supplementary Fig. 2c, left). This was validated by quantifying ARID2 immunofluorescence from control and mutant metaphase-I spermatocytes (Supplementary Fig. 2c, right). At the same time, PLK1 levels were also significantly reduced in mutant metaphase-I spermatocytes relative to controls (Supplementary Fig. 2c, right). To understand the underlying cause, we examined *Plk1* mRNA abundance in response to the loss of ARID2 by qRT-PCR. Since *Plk1* transcript levels have been shown to increase during the first wave of spermatogenesis,

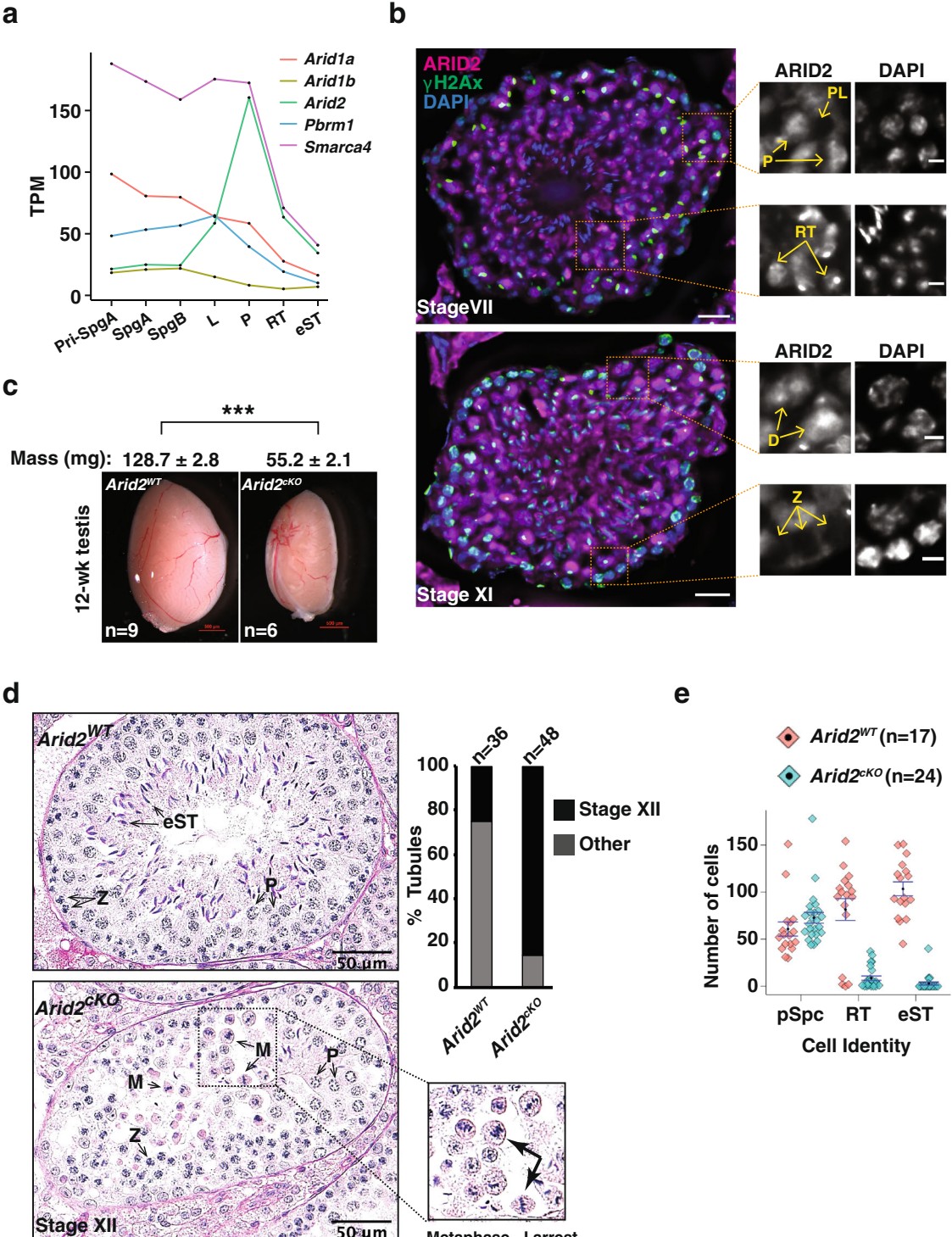

**Fig. 1 The loss of ARID2 results in late meiosis-I arrest. a** mRNA expression profile of SWI/SNF subunits in purified germ cell populations[13]. **b** Testis cryosections displaying Stage VII (top) and XI (bottom) seminiferous tubules immunolabelled for ARID2 (magenta), γH2Ax (green), and counterstained with DAPI (blue), scale bar: 20 μm. Magnified panels display distinct spermatogenic cell types, scale bar: 5 μm. Magnification: 40x. Immunostainings were performed twice on testis cryosection obtained from two adult males. **c** *Arid2^WT* and *Arid2^cKO* testis mounts. Average testis mass (mg) ± standard error (SEM) obtained from multiple measurements (*n*) are indicated. ***$p = 8.9 \times 10^{-9}$ was calculated by a two-tailed unpaired Student's *t*-test. Scale bar: 500 μm. **d** PAS-stained testis section (left) and proportion of stage XII tubules (right) from 12-week-old *Arid2^WT* and *Arid2^cKO* testis. A magnified view of *Arid2^cKO* metaphase spermatocytes is displayed. Scale bar: 50 μm, magnification: 40x. **e** Number of prophase-I spermatocytes (pSpc) and spermatid cell populations quantified from 12-week-old *Arid2^WT* and *Arid2^cKO* PAS-stained testes sections and expressed as mean ± SEM (standard error of measurement) (**d**, **e**). The number of tubules (*n*) examined for each genotype are indicated (**a**, **b**, **d**). Pri-SpgA primitive spermatogonia type A, SpgA spermatogonia type A, SpgB spermatogonia type B, PL preleptotene spermatocyte, L leptotene spermatocyte, P pachytene spermatocyte, Z zygotene spermatocyte, M metaphase-I spermatocyte, RT round spermatid, and eST elongated spermatid.

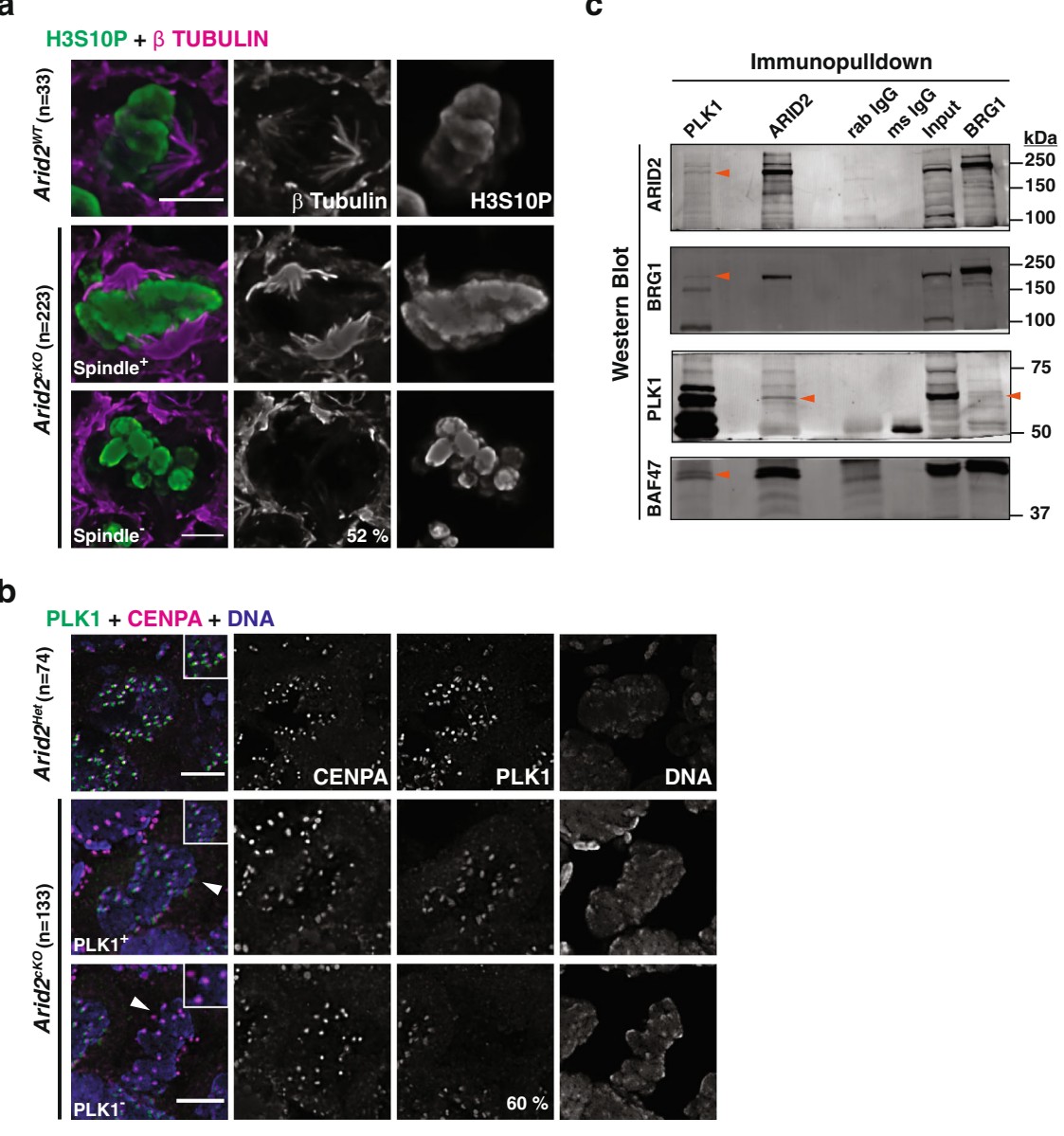

**Fig. 2 ARID2 influences spindle assembly and PLK1 association with kinetochore. a**, **b** Metaphase-I spermatocytes from control and *Arid2^cKO* testes cryosections immunolabelled for: **a** H3S10P (green) and β-Tubulin (magenta), **b** PLK1 (green), CENPA (magenta), and counterstained with DAPI (blue). Arrowheads label metaphase-I spermatocytes and panel insets highlight centromeres. **a**, **b** Scale bars: 5 μm, magnification: 100x. Internal control (Spindle ⁺/PLK1⁺) and mutant metaphase-I spermatocytes (Spindle⁻/PLK1⁻) were imaged from the same section. Number (*n*) of metaphase-I spermatocytes scored and proportion (%) of mutant cells are indicated. **c** PLK1, ARID2, and BRG1 Co-IPs from spermatocyte protein extracts. Immunoblotted proteins are labeled with orange arrowheads. ARID2 and BRG1 Co-IPs were reproducibly repeated three times along with two PLK1 Co-IPs.

peaking at the end of meiosis-I[28], we quantified mRNA abundance in spermatogenic cells isolated from control and *Arid2^cKO* testes at postnatal day P23, a stage that coincides with the normal appearance of late prophase-I (diplonema) and metaphase-I spermatocytes. At P23, *Arid2^cKO* testes displayed significantly lower *Plk1* mRNA levels relative to control testes (Supplementary Fig. 2d). Despite reduced *Plk1* mRNA levels, the corresponding protein levels appeared unchanged in *Arid2^cKO* diplotene spermatocytes, which were identified by the presence of de-synapsing axial elements marked with HORMAD1[29] (Supplementary Fig. 2e). Given that spermatogenic transcription is known to peak at the end of prophase-I[15,30], it is reasonable to propose that *Plk1* repression in *Arid2^cKO* diplotene spermatocytes underlies the depletion and therefore kinetochore disassociation of PLK1 at the onset of metaphase-I. Interestingly, we also identified a physical

interaction between PLK1 and PBAF by performing co-immunoprecipitations (Co-IP) on protein extracts isolated from spermatocyte enriched populations harvested at P19-P20. We detected ARID2, BRG1, and BAF47(SMARCB1/SNF5), a core SWI/SNF subunit, co-immunoprecipitating with PLK1 (Fig. 2c). Similarly, PLK1 was detected in ARID2 and BRG1 Co-IPs (Fig. 2c), thereby confirming an association between PLK1 and PBAF in spermatocytes. These data suggest that PBAF localizes to centromeric regions.

**ARID2 governs the targeting of H3T3P and H2AT120P to centromeres.** Given that ARID2 interacts with PLK1, which is known to influence centromeric chromatin during mitosis[31], we were curious to determine the influence of ARID2 on histone modifications that regulate cell division. These include histone H3

threonine3 phosphorylation (H3T3P) and histone H2A threonine120 phosphorylation (H2AT120P), which are concomitantly deposited by HASPIN kinase[32,33] and spindle checkpoint kinase BUB1[34], respectively. Interestingly, perturbation of H3T3P levels are detrimental to the completion of meiosis-I in mouse oocytes[33]. Therefore, we first monitored H3T3P in $Arid2^{WT}$ (normal) metaphase-I spermatocyte squashes. Here, H3T3P overlapped with centromeres and was distributed along the inter-chromatid axes (ICA) (Fig. 3a). In $Arid2^{cKO}$ squashes, the absence of centromeric PLK1 was used to differentiate mutant metaphase-I spermatocytes from internal controls (Fig. 2b and Supplementary Fig. 2c). H3T3P distribution was limited in internal controls (10%), resembling normal metaphase-I spermatocytes (Fig. 3a). In contrast to normal spermatocytes and internal controls, H3T3P was spread chromosome-wide in most mutant metaphase-I spermatocytes (60%), with fewer mutants (30%) displaying low levels of chromatid-bound H3T3P (Fig. 3a). From western blot analyses, total H3T3P levels appeared stable upon the loss of ARID2 in testis histone extracts obtained at P23 and P27. Therefore, the chromosome-wide distribution of H3T3P is unlikely to result from altered HASPIN activity in the mutants (Supplementary Fig. 3a, top panel).

Next, we determined the influence of ARID2 on H2AT120P. During mitosis, H2AT120P is restricted to centromeres[34,35]. While control ($Arid2^{Het}$) metaphase-I spermatocytes and internal controls ($Arid2^{cKO}$) displayed H2AT120P limited to centromeres, mutant metaphase-I spermatocytes (62%) exhibited broad chromosomal distribution of H2AT120P (Fig. 3b). The loss of ARID2 did not alter total H2AT120P abundance, suggesting normal BUB1 activity in the mutant (Supplementary Fig. 3b). Therefore, ARID2 is required to restrict H3T3P and H2AT120P to centromeric regions in spermatocytes.

The ectopic distribution of centromeric histone marks led us to investigate the impact of ARID2 depletion on chromatin organization during metaphase-I. To do this, we treated $Arid2^{WT}$ and $Arid2^{cKO}$ testes cryosections with varying concentrations of MNase (0–200 Kunitz units) and monitored chromatin digestion in normal and mutant metaphase-I spermatocytes, identified by PLK1 and ARID2 IF (Fig. 3c). Susceptibility to MNase digestion was qualitatively assessed by DAPI staining, which provided a measure of accessibility. The metaphase plate appeared shredded in $Arid2^{WT}$ spermatocytes treated with 20–200 units MNase, which is indicative of permissive chromatin (Fig. 3c). Consistent with its heterochromatic nature, centromere-proximal regions marked with PLK1 appeared resistant to MNase digestion (Fig. 3c). We observed a dose-response to MNase treatment, with the proportion of normal metaphase-I spermatocytes exhibiting chromatin digestion increasing from 16 to 79% over a 100-fold concentration (2–200 Kunitz units) (Fig. 3d). Similar MNase sensitivity was observed for internal controls from $Arid2^{cKO}$ cryosections (Fig. 3c, d). In contrast, chromatin from mutants in $Arid2^{cKO}$ cryosections appeared highly inaccessible, evidenced by the lack of DNA digestion even upon treatment with 200 Kunitz units of MNase (Fig. 3c). Consistent with reduced chromatin accessibility in the absence of ARID2, we identified mis-localization of H3K9me3, a histone modification normally associated with pericentric heterochromatin[36]. Many mutant metaphase-I spermatocytes (PLK1⁻) displayed chromosome-wide H3K9me3 (38%) relative to internal controls and control ($Arid2^{Het}$) metaphase-I spermatocytes (Supplementary Fig. 3c), without an increase in total H3K9me3 abundance (Supplementary Fig. 3a, bottom panel). Therefore, ARID2 promotes chromatin accessibility at metaphase-I.

chromatin structure in the absence of ARID2 prompted us to determine the genomic associations of the PBAF complex during metaphase-I. Given that ARID2 levels peak during pachynema (Fig. 1a, b), we performed CUT&RUN (Cleavage Under Targets & Release Using Nuclease) in duplicates to map ARID2 and BRG1 binding sites in spermatogenic populations enriched for late prophase-I (pachynema/diplonema) and metaphase spermatocytes. Since metaphase-I spermatocytes constitute a mere 4% of total spermatocytes in the developing testis[37], we increased the proportion of metaphase spermatocytes by synchronizing spermatogenesis. This is accomplished by temporarily inhibiting retinoic acid (RA) synthesis in mice, which has been used previously to enrich specific germ cell populations[38,39]. Here, juvenile mice were treated from P1–10 with WIN 18,466, a potent and selective inhibitor of RA synthesis. As a result, there was an accumulation of undifferentiated type A spermatogonia (SpgA). At P11, a single dose of exogenous RA initiates SpgA differentiation, culminating in synchronous spermatogenesis (Fig. 4a). In synchronized testes, we detected accumulation of pachytene/diplotene and more notably a higher proportion (14–16%) of spermatocytes at metaphase at P28 (17 days after RA treatment) and P28.5 (17.5 days after RA treatment), respectively (Supplementary Fig. 4a). There were increased numbers of round spermatids (RT) in P28.5 relative to P28 seminiferous tubules, highlighting the rapidity of reductional meiosis. Additionally, the proportions of leptonema/zygonema spermatocytes were markedly decreased at P28.5. The near absence of ARID2 in early prophase-I spermatocytes (Fig. 1b) suggests that potential sites mapped by CUT&RUN would be representative of its genomic associations late in meiosis-I.

We examined the enrichment of ARID2 relative to a no IgG control and performed k-means clustering to identify potential target TSSs at P28 and P28.5. ARID2-occupied TSSs were grouped in cluster-1 (Cl-1), while cluster-2 (Cl-2) represented TSSs lacking ARID2 (Fig. 4b). Like ARID2, BRG1 was also detected at TSSs associated with Cl-1 at P28 and P28.5. To validate our results we analyzed previously generated ChIP-seq data from P18 testes (without synchronization)[6], that are normally enriched for pachytene spermatocytes. By monitoring P18 BRG1 ChIP enrichment at Cl-1 and Cl-2 (from CUT&RUN), we observed robust BRG1 occupancy at TSSs associated with candidate PBAF targets (Cl-1), as early as pachynema (Supplementary Fig. 4b). Additionally, some Cl-2 associated TSSs also displayed moderate BRG1 occupancy when detected by ChIP-seq. It is possible that these Cl-2 sites went undetected by CUT&RUN (Fig. 4b) owing to differences in sensitivity compared to ChIP-seq or the cellular profiles assayed. Therefore, Cl-1 TSSs represent PBAF target genes late in meiosis-I. Consistent with a role in transcriptional regulation, PBAF occupancy near $Plk1$ promoters was indicated by ARID2 and BRG1 enrichment and further validated by calling ARID2 peaks using SEACR (Span Enrichment Analysis for CUT&RUN) at P28[40] (Supplementary Fig. 4c).

In addition to gene promoters, we also identified ARID2 and BRG1 associations with centromeric regions during meiosis. We performed ChIP on spermatogenic cells isolated from P19 testes, which are developmentally enriched for pachytene and diplotene spermatocytes. Both ARID2 and BRG1 were enriched at centromeric minisatellites, comparable to known centromeric marker, H3T3P (Fig. 4c). H3K27me3, a mark usually associated with euchromatic gene promoters (Fig. 4c, $Gdnf$), was used as a negative control. As expected, both centric and pericentric satellites appeared depleted for H3K37me3. In contrast to minisatellites, pericentric major satellites displayed poor ARID2 and BRG1 enrichment.

**Genomic associations reveal PBAF enrichment at gene promoters and centromere.** The mis-regulation of $Plk1$ and changes in

**ARID2 associated genes regulate spindle assembly and chromosome segregation.** To gain insight into the functions of genes

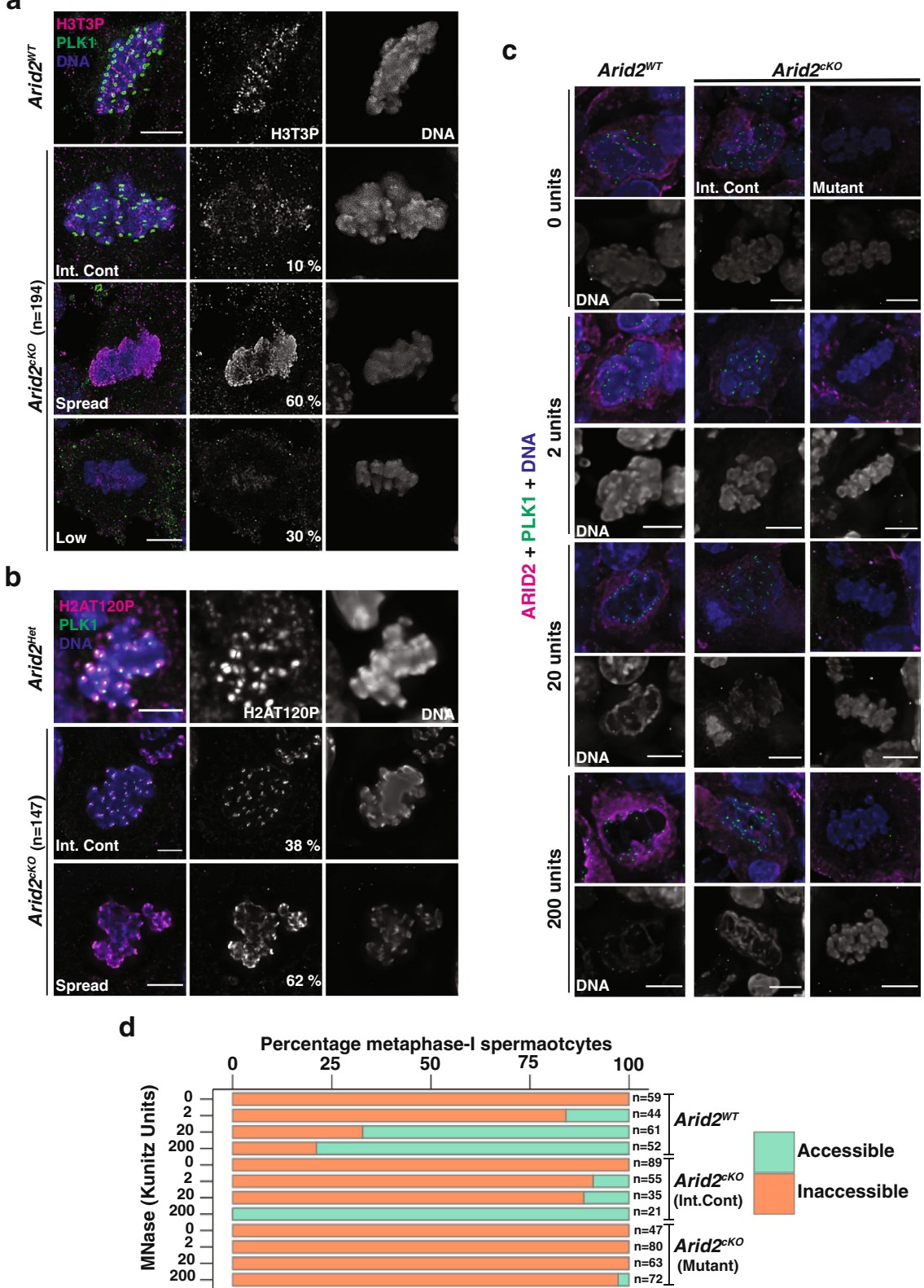

**Fig. 3 ARID2 regulates chromatin organization in metaphase-I spermatocytes. a**, **b** Control and *Arid2cKO* metaphase-I spermatocyte squashes immunolabelled for PLK1 (green) and **a** H3T3P (magenta) and **b** H2AT120P (magenta). **c**, **d** Control and *Arid2cKO* metaphase-I spermatocytes from cryosections treated with MNase (0–200 Kunitz units) **c** immunolabelled for PLK1 (green) and ARID2 (magenta), and **d** scored for chromatin accessibility indicated by the proportion of cells (horizontal axis) susceptible (accessible) or resistant (inaccessible) to MNase digestion at indicated MNase concentrations (vertical axis). The number of spermatocytes scored (*n*) are indicated. **a**–**c** DNA stained with DAPI. Scale bar: 5 μm, magnification: 100.8x. Int.Control: Internal control.

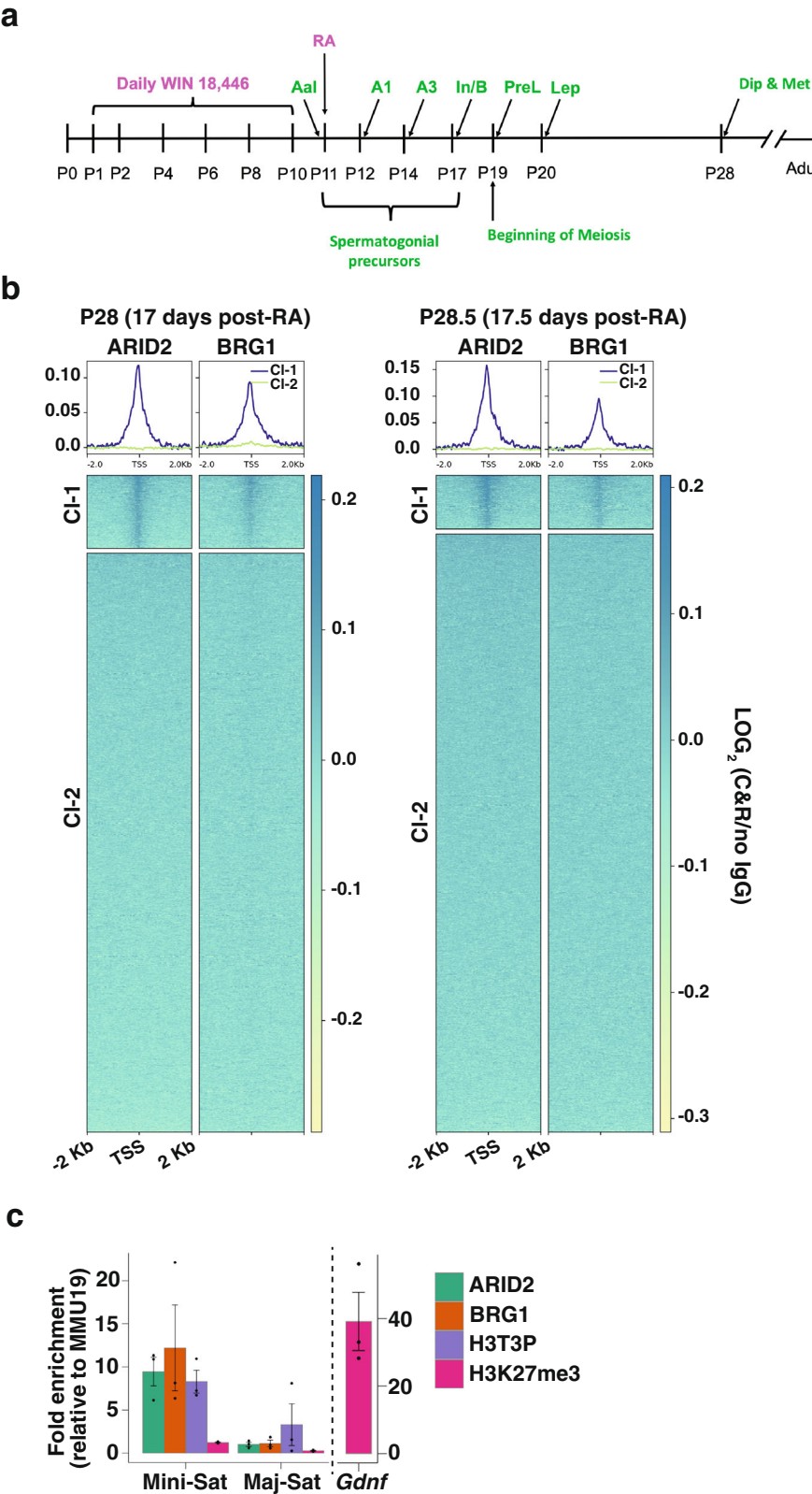

**Fig. 4 Genome-wide associations of PBAF in spermatocytes. a** Schematic illustrating protocol used to synchronize spermatogenesis using WIN 18,446 and retinoic acid (RA), **b** Heatmap displaying the average enrichment of ARID2 and BRG1 relative to no IgG control from spermatocyte populations isolated at P28 and P28.5. Enrichment is displayed over a 4 Kb window centered at transcription start sites (TSS's) of RefSeq genes. K-means clusters (Cl-1 and 2) are based on ARID2 enrichment. **c** ChIP-qPCR analysis of ARID2, BRG1, H3T3P, and H3K27me3 enrichment at centromeric minisatellites (Mini-Sat) and pericentric major satellites (Major-Sat) repeats. Fold enrichment (y-axis) were calculated relative to known gene desert (MMU19) and were expressed as mean ± SEM. Dots (black) represent values obtained from three independent experiments.

associated with PBAF, we analyzed Cl-1-associated regions at P28 and P28.5 using Genomic Regions Enrichment of Annotations Tools (GREAT)[41]. GREAT analysis revealed a striking enrichment for gene ontology (GO) terms associated with processes such as spindle assembly and nuclear division (Supplementary Fig. 5a, b). Additionally, many PBAF target genes were associated with components of centrosome and kinetochore (Supplementary Fig. 5a, b). These GO terms are particularly relevant to the *Arid2cKO* phenotype and suggest that the mis-regulated transcription of key target genes might underlie the metaphase-I arrest.

**ARID2 influences centrosome formation and activity**. Notably, ARID2 associated genes encode centrosomal proteins such as γ-Tub/TUBG1 (γ-Tubulin) and AURKA (Aurora kinase A), which are expressed during meiosis and mitosis (Supplementary Fig. 6a). These candidate factors are essential for centrosome maturation, a process central to bipolar spindle assembly[42–45]. We reasoned that misexpression of associated genes would underlie the spindle defects seen in *Arid2cKO* spermatocytes (Fig. 2a). By qRT-PCR, no significant differences in mRNA abundance of either *Tubg1* or *Aurka* were detected in *Arid2cKO* relative to control (*Arid2Het*) spermatocyte enriched populations (Supplementary Fig. 7a, left panel). Since mRNA levels are not always predictive of protein levels, we monitored the abundance of γ-Tubulin and AURKA, in metaphase-I spermatocytes by IF.

γ-Tubulin is a key constituent of centrosomes in mitotic cells and acentriolar microtubule organizing centers (aMTOCs) in oocytes[46]. Like mitotic cells, metaphase spermatocytes display centrosomes occupying spindle poles[47]. Consistent with this, we observed two prominent foci labeled with γ-Tubulin in control (*Arid2Het*) metaphase-I spermatocytes (Fig. 5a). In contrast, *Arid2cKO* metaphase-I spermatocytes displayed diffuse puncta or multiple foci of γ-Tubulin (Fig. 5a). These data suggest that ARID2 is necessary to maintain centrosome integrity.

Next, we monitored the expression and localization of AURKA. In spermatocytes, AURKA localization overlaps the spindle poles[48]. We observed a similar cytosolic enrichment around the spindle poles in control (*Arid2Het*) and internal control metaphase-I spermatocytes (Supplementary Fig. 8a). In contrast, AURKA appeared displaced, spreading diffusely across the metaphase plate in mutant metaphase-I spermatocytes exhibiting spindle defects (Supplementary Fig. 8a, row 3–6). Furthermore, AURKA abundance was significantly elevated in *Arid2cKO* relative to control metaphase-I spermatocytes (Supplementary Fig. 8b). While spindle fibers appeared deficient in most mutant metaphase-I spermatocytes (Supplementary Fig. 8a, row 3–5), a minority displayed multipolar spindles (Supplementary Fig. 8a, row 6), a phenotype that had escaped previous characterization (Fig. 2a). These phenotypes are consistent with previously reported spindle defects observed upon the overexpression of *Aurka* in oocytes[49]. Therefore, defects in centrosome formation and AURKA regulation compromise spindle assembly in *Arid2cKO* metaphase-I spermatocytes.

Since the activation of AURKA by auto-phosphorylation at threonine 288 (pT288-AURKA) signals its competence in spindle assembly, we determined the influence of ARID2 on pT288-AURKA. At metaphase, pT288-AURKA has been shown to localize to the centrosomes and aMTOCs[43]. Expectedly, we observed a striking accumulation of pAURKA-T288 at spindle poles in control (*Arid2Het*) and internal control metaphase-I spermatocytes (Fig. 5b). Surprisingly, pT288-AURKA signals were also detected near microtubule plus (+) ends, overlapping chromosomes. Unlike spermatocytes, mitotic cells from control testes mostly displayed centrosomal pT288-AURKA, suggesting

that its chromosomal localization was meiosis-specific (Supplementary Fig. 8c).

Furthermore, pT288-AURKA localization to metaphase-I chromosomes was reproducibly detected using another antibody (pT288-AURKA[CST], Supplementary Fig. 8d). While these data are consistent with a previous report from oocytes[49], we cannot rule out the possibility that the pT288-AURKA antibodies might recognize chromatin-bound meiotic kinases like AURKC (Aurora kinase C). Therefore, we limited our analysis to determine the influence of ARID2 on the centrosomal accumulation of pT288-AURKA. We observed a striking perturbation of pT288-AURKA accumulation at the spindle poles of *Arid2cKO* metaphase-I spermatocytes. Here, pT288-AURKA appeared displaced and reduced overall (Fig. 5b and Supplementary Fig. 8b). Therefore, AURKA activation is deficient upon the loss of ARID2.

**ARID2 regulates the expression of factors essential for AURKA activity**. AURKA activity and its localization is governed by its association with cofactors[43]. Therefore, we hypothesized that the defects in AURKA localization and activation might result from the mis-expression of known cofactors in the absence of ARID2. Intriguingly, genes encoding BORA and TPX2, two known AURKA binding partners, that also influence spindle assembly, were identified as ARID2 targets by CUT&RUN (Supplementary Fig. 6). Only *Bora* mRNA levels were significantly reduced by 42% in *Arid2cKO* relative to control spermatocyte populations obtained at P23 (Supplementary Fig. 7a, left panel). Consistent with a transcriptional defect, BORA protein levels were significantly reduced by 63% in the absence of ARID2 (Supplementary Fig. 7b). Unsurprisingly, by IF, BORA levels were reduced in most mutants (83%) and appeared diffusely spread in the absence of ARID2. (Fig. 5c). Therefore, a deficiency in BORA might result in the displacement of AURKA. Interestingly, in oocytes, the inhibition or knockdown of BORA has been shown to result in the loss of AURKA[50]. We were, therefore, curious to understand why the levels of AURKA remained elevated upon the loss of ARID2 in spermatocytes (Supplementary Fig. 8b). To address this, we examined PBAF target genes, focusing on candidates known to regulate AURKA degradation. We identified targets that constitute core subunits of the anaphase-promoting complex (APC), a known E3 ubiquitin ligase[51] (Supplementary Fig. 6). During metaphase, AURKA proteolysis is known to occur via the concerted activity of the APC bound to its coactivator, FZR1 (Fizzy-related or CDH1)[52]. Since this interaction is mediated by the APC scaffolding subunit, CDC27 (APC3), we determined its expression in response to the loss of ARID2. While *Cdc27* mRNA abundance was significantly reduced by 50% in *Arid2cKO* relative *Arid2Het* spermatocyte enriched populations at P23, total CDC27 protein levels remained unchanged as detected by western blotting (Supplementary Fig. 7a, b). However, a careful examination of CDC27 abundance at metaphase-I by IF revealed a significant decrease in CDC27 levels in mutant relative control spermatocytes (Supplementary Fig. 9a, b). Therefore, potential deficiencies in APC activity might result in the accumulation of AURKA upon ARID2 loss. More interestingly, the loss of APC activity represents yet another feature of *Arid2cKO* metaphase-I arrested spermatocytes.

**CPC localization is perturbed in the absence of ARID2**. Next, we shifted our attention to other aurora kinases essential for meiotic cell division. These include the chromosome passenger complex (CPC) kinases AURKB (Aurora kinase B) and AURKC (Aurora kinase C, meiosis-specific)[53], which are normally chromatin-bound[54–56]. We identified ARID2 peaks associated with promoters of both *Aurkc* and *Aurkb* at P28 (Supplementary

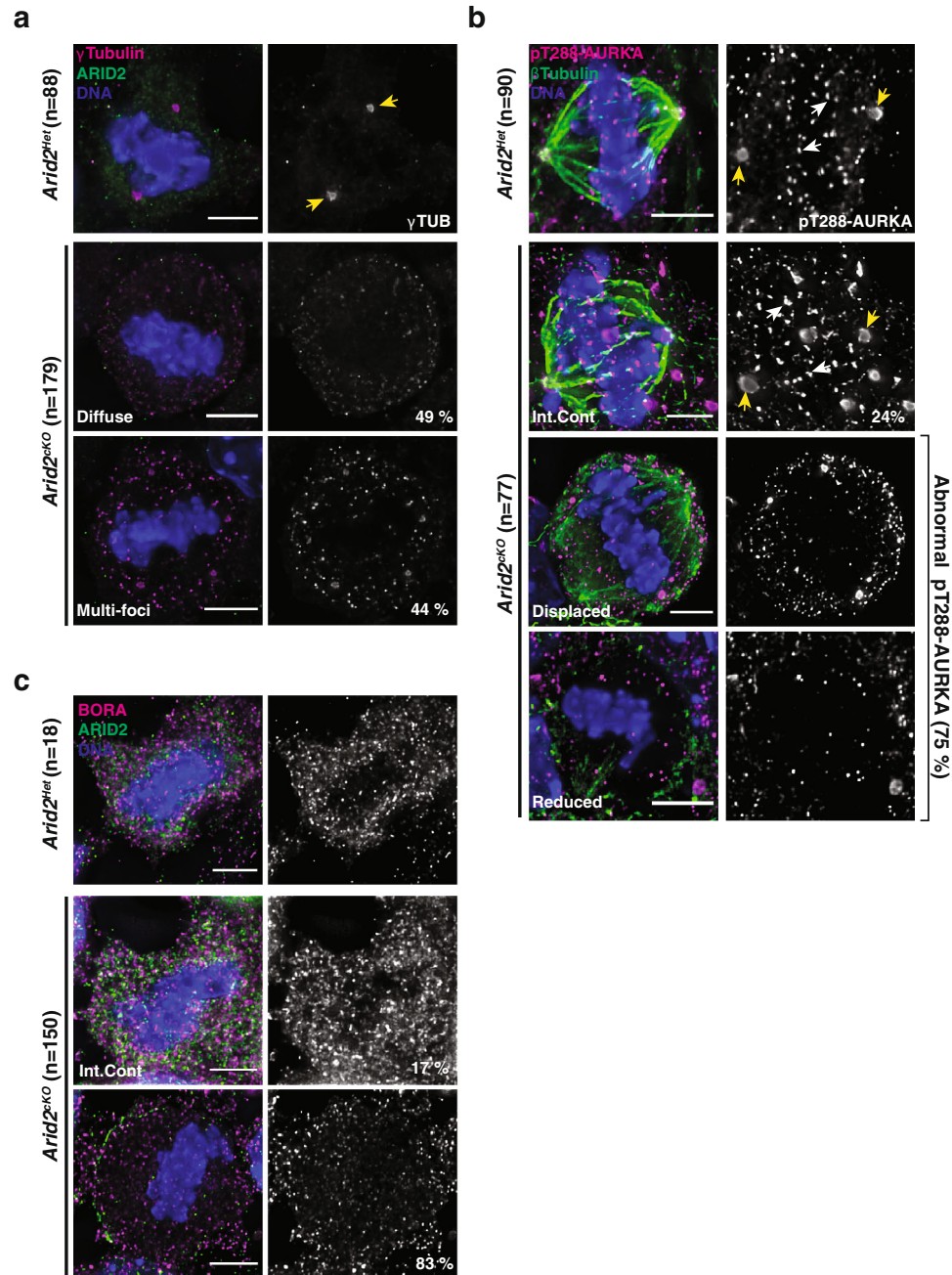

**Fig. 5 ARID2 influences the localization of centrosome-associated factors.** *Arid2^Het* and *Arid2^cKO* metaphase-I spermatocytes immunolabelled for **a** γ-Tubulin (magenta) and ARID2 (green), **b** pT288-AURKA (magenta) and β-Tubulin (green), **c** BORA (magenta) and ARID2 (green), **a–c** DNA counterstained with DAPI (blue). **a, b** Centrosomes (yellow arrowheads) and chromosomal loci (white arrowheads) are indicated. Scale bars: 5 μm, magnification:100.8x. Number (*n*) of metaphase-I spermatocytes scored and proportion (%) of abnormal cells are indicated.

Fig. 6). Furthermore, *Aurkc* mRNA levels appeared significantly depleted in *Arid2^cKO* relative to control spermatocyte enriched populations (Supplementary Fig. 7a, right panel). In contrast, we only detected a moderate decrease in *Aurkb* mRNA at P23, upon the loss of ARID2 (Supplementary Fig. 7a, right panel). These transcriptional changes were reflected in total AURKC and AURKB protein levels which were reduced by 72 and 22% respectively (Supplementary Fig. 7b).

Consistent with its mis-repression, AURKC was undetectable by IF in mutant metaphase-I spermatocytes (71%) relative to internal controls (26%) and controls where it localizes to chromatin (Fig. 6a). Next, we determined whether the loss of AURKC impacted its activity. We monitored the phosphorylation

of INCENP (inner centromere protein), a known substrate of AURKC during meiosis[57]. Similar to previous studies[55], phosphorylated INCENP (pINCENP) was detected at centromeres and along the ICA (inter-chromatid axis) in normal metaphase-I spermatocytes and internal controls from *Arid2^cKO* testis. In contrast, chromatin-bound pINCENP was undetectable in mutant metaphase-I spermatocytes (87%), (Fig. 6b). Therefore, CPC activity is compromised in the absence of ARID2.

Since the loss of AURKC can be partially compensated by AURKB during female meiosis[57], we examined AURKB localization in the absence of ARID2. In normal metaphase-I spermatocytes, AURKB puncta were enriched on chromatin (Fig. 6c). This was also true in internal controls (29%) that

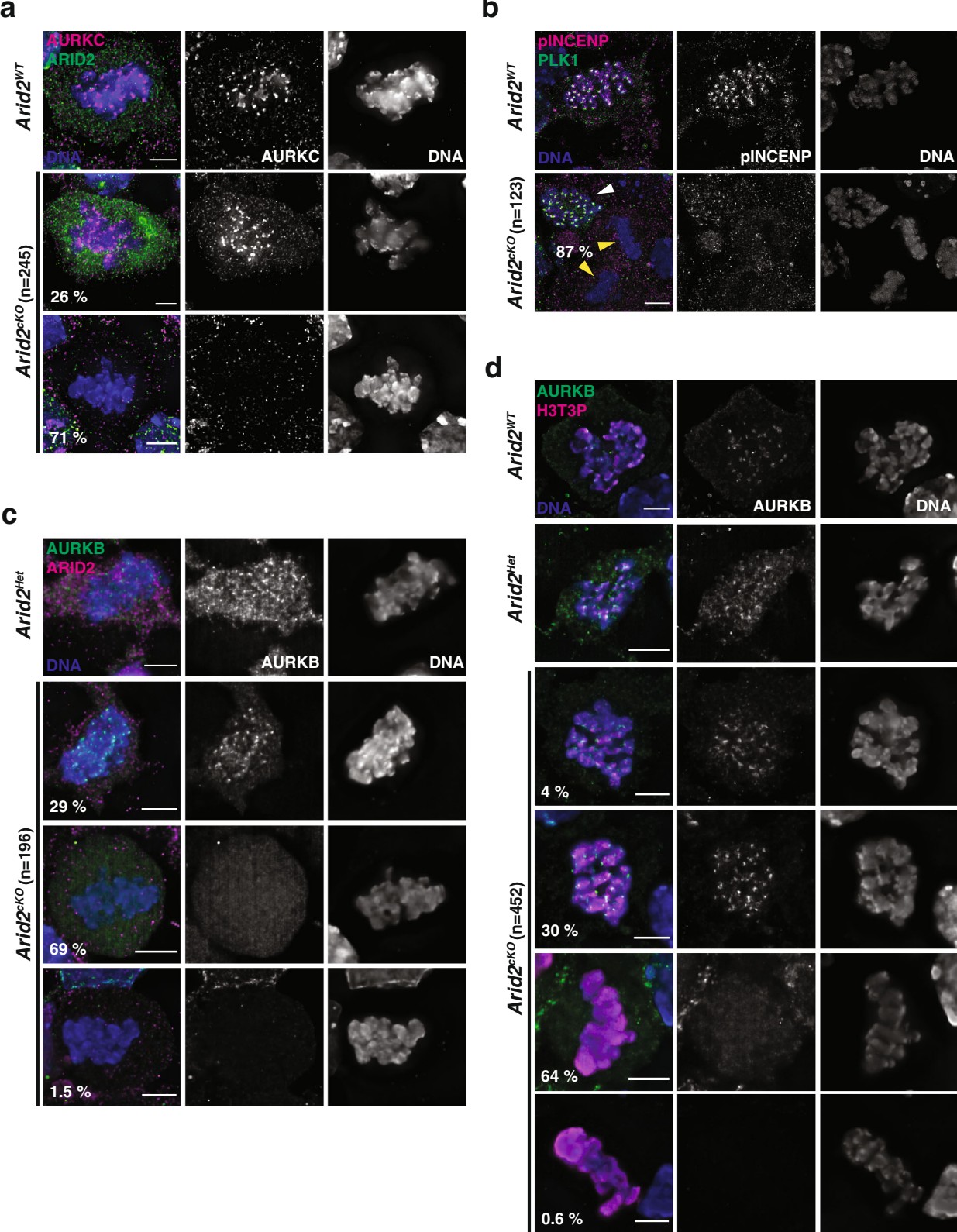

**Fig. 6 ARID2 influences chromosome passenger complex recruitment. a, b** *Arid2^WT* and *Arid2^cKO* metaphase-I spermatocyte squashes immunolabelled for **a** AURKC (magenta) and ARID2 (green), **b** pINCENP (magenta) and PLK1 (green). Internal controls (white arrowhead) and mutants (yellow arrowhead) are indicated. **c, d** Control and *Arid2^cKO* metaphase-I spermatocyte squashes immunolabelled for AURKB (green) and **c** ARID2 (magenta), **d** H3T3P (magenta). **a–d** DNA stained with DAPI (blue), number of metaphase-I spermatocytes scored (*n*), and their proportions (%) in *Arid2^cKO* squashes are indicated. Scale bar: 5 μm, magnification: 100.8x.

displayed a clear chromatin association (Fig. 6c, row 2). In contrast, chromosomal AURKB puncta were undetectable in ARID2 deficient metaphase-I spermatocytes, with most mutants displaying a diffuse localization (69%) or a striking loss (1.5%) of AURKB (Fig. 6c, row 3, 4). The destabilization of AURKB association with chromatin might be a consequence of the moderate AURKB turnover in mutant metaphase-I (Supplementary Fig. 7b).

These results are surprising given that H3T3P and H2AT120P, known signals of CPC recruitment[33,34,58–60], are distributed chromosome-wide in the absence of ARID2 (Fig. 3a, b). We, therefore, confirmed the defect in CPC localization by co-immunostaining for AURKB and H3T3P. We observed punctate enrichment of AURKB near regions marked by H3T3P in normal metaphase-I spermatocytes (Fig. 6d, row 1,2) and internal controls from *Arid2cKO* testis (Fig. 6d, row 3, 4). Consistent with our earlier observations, AURKB was diffusely spread in most mutant metaphase-I spermatocytes displaying chromosome-wide H3T3P (Fig. 6d, row 5, 6). Our data indicate that *Arid2cKO* metaphase-I spermatocytes lack a functional CPC that can bind chromatin. Therefore, spermatocytes are unable to exit metaphase-I.

**ARID2 influences the function of CDCA2-PP1γ.** The presence of ectopic H3T3P and H2AT120P indicates that *Arid2cKO* metaphase-I spermatocytes are unable to erase these histone modifications. Interestingly, PBAF is associated with genes encoding factors known to limit H3T3P during mitosis. These include *Cdca2* (cell division cycle associated 2) better known as *Repo-man*, and *Ppp1cc* (protein phosphatase 1 catalytic subunit gamma, *Pp1γ*) (Supplementary Fig. 6). During mitosis, H3T3P is initially deposited chromosome-wide and then subsequently targeted to centromeres through the activity of the CDCA2-PP1 complex, where CDCA2 serves as a docking module to recruit PP1γ phosphatase to chromatin[60,61]. Although the meiotic roles of CDCA2 and PP1γ are unknown, by RNA-seq[13] their cognate transcript levels appear elevated during the transition from pachynema to spermiogenesis (Supplementary Fig. 10a). We, therefore, decided to monitor CDCA2 and PP1γ localization in metaphase-I spermatocytes by IF. Control (*Arid2Het*) metaphase-I spermatocytes exhibited both chromatin-associated and cytosolic pools of CDCA2 (Supplementary Fig. 10b). In contrast, *Arid2cKO* metaphase-I spermatocytes frequently displayed a loss of CDCA2 (51%) or reduced abundance (4%) relative to the controls (Supplementary Fig. 10b). Interestingly, *Cdca2* mRNA abundance was reduced by 30% in *Arid2cKO* relative to *Arid2Het* spermatogenic cells isolated at P23 (Supplementary Fig. 7b, right panel). Despite these reduced mRNA levels, we were able to identify mutant metaphase-I spermatocytes (26%) with CDCA2 diffusely associated with chromatin (Supplementary Fig. 10b, row 4). Intriguingly, perturbations in CDCA2 levels correlated with the mis-targeting of H3T3P in *Arid2cKO* metaphase-I spermatocytes (Fig. 3a).

To confirm this, we performed direct immunofluorescence to monitor CDCA2 and H3T3P in metaphase-I spermatocytes. Most control metaphase-I spermatocytes (89%) with limited H3T3P displayed cytosolic and chromatin-associated CDCA2, much like the typical pattern observed earlier (Fig. 7a, row 1, Supplementary Fig. 10b, row 1). Additionally, a small proportion of control metaphase-I spermatocytes (5%) displayed chromosome-wide H3T3P, coincident with a diffuse CDCA2 signal that appeared delocalized (Fig. 7a, row 2). These cells likely represent a stage preceding the removal of H3T3P along chromosomes arms. In contrast to controls, fewer *Arid2cKO* metaphase-I spermatocytes appeared typical (19%) and had limited H3T3P and normal

CDCA2, likely representing internal controls (Fig. 7a, row 3). The remaining metaphase-I spermatocytes appeared abnormal, falling into two distinct categories: (1) mutants displaying either a striking loss of CDCA2 (24%); or (2) a moderate depletion of CDCA2 with residual chromatin association (42%) (Fig. 7a, row 4–6). While the depletion of CDCA2 accounts for the failure to limit H3T3P, the detection of chromatin-bound CDCA2 in many *Arid2cKO* metaphase-I spermatocytes displaying ectopic H3T3P (Fig. 7a, row 6) is counterintuitive. It is possible that a deficiency in PP1γ results in the failure to restrict H3T3P upon ARID2 loss. In fact, *Pp1γ* mRNA abundance is significantly reduced by 40% in *Arid2cKO* relative to control spermatocyte enriched populations (Supplementary Fig. 7a, right panel). To monitor corresponding changes in protein levels, we performed PP1γ IF. By simultaneously staining for H3T3P, we directly correlated changes to PP1γ. Most control metaphase-I spermatocytes (94%) displayed limited H3T3P with PP1γ localized to the cytosol, and chromatin to a lesser extent (Fig. 7b, row 1). Comparatively, few controls displayed chromosome-wide H3T3P accompanied by diffuse (3%) or depleted (2%) PP1γ. These data show that PP1γ is localized similarly to its binding partner CDCA2 during metaphase-I (Fig. 7a). Compared to controls, the proportion of *Arid2cKO* metaphase-I spermatocytes displaying typical PP1γ localization pattern (29%) was reduced significantly, while the percentage of those displaying ectopic H3T3P along with diffuse PP1γ (4%) remained steady (Fig. 7b, row 4, 5). Remarkably, the majority (66%) of *Arid2cKO* metaphase-I spermatocytes displayed reduced PP1γ abundance relative to the controls (Fig. 7b, row 6). This deficiency was further validated by quantifying PP1γ fluorescence; its intensity was significantly reduced in the majority of *Arid2cKO*, relative to control metaphase-I spermatocytes (Supplementary Fig. 10c). We also quantified H3T3P fluorescence from these cells and identified discordant changes relative to PP1γ in control and *Arid2cKO* metaphase-I spermatocytes (Supplementary Fig. 10c). Therefore, the failure to limit H3T3P during metaphase-I can be explained by a lack of PP1γ in the absence of ARID2.

**PP2A stability is compromised in the absence of ARID2.** In addition to *Pp1γ*, we also identified ARID2 associations at *Ppp2r1a* (protein phosphatase 2 regulatory subunit A alpha) and *Ppp2ca* (protein phosphatase 2 catalytic subunit alpha isoform) promoters (Supplementary Fig. 6). PPP2R1A and PPP2CA constitute the scaffolding subunit and catalytic subunit respectively, of the promitotic PP2A (protein phosphatase 2 A) complex[62]. Intriguingly, we had previously identified PPP2R1A (three unique peptides with 9.3% coverage) in a proteomic analysis of BRG1 interacting proteins in spermatocytes[6]. To determine whether PPP2R1A is associated with PBAF, we performed ARID2 and PPP2R1A Co-IP. We confirmed the presence of PPP2R1A in ARID2 Co-IP (Fig. 8a, top) and detected ARID2, BRG1, and BAF47 immunoprecipitating with PPP2R1A (Fig. 8a, bottom). This association prompted us to examine PP2A localization in metaphase-I spermatocytes in response to the loss of ARID2. First, we monitored PPP2R1A which was detected on chromatin and cytosolically in control metaphase-I spermatocytes and internal controls from *Arid2cKO* testes (Fig. 8b, row 1,2). In contrast, PPP2R1A appeared depleted in mutant metaphase-I spermatocytes (Fig. 8b, row 3, 4). Next, we examined PPP2CA, which overlapped ARID2 in control metaphase-I spermatocytes and internal controls from *Arid2cKO* testes (Fig. 8c, row 1, 2), but was diminished or absent in mutant metaphase-I spermatocytes (Fig. 8c, row 3, 4). Interestingly, by qRT-PCR, we were unable to detect a decrease in the mRNA abundances of *Ppp2r1a* and *Ppp2ca* in response to the loss of ARID2 (Supplementary Fig. 7a,

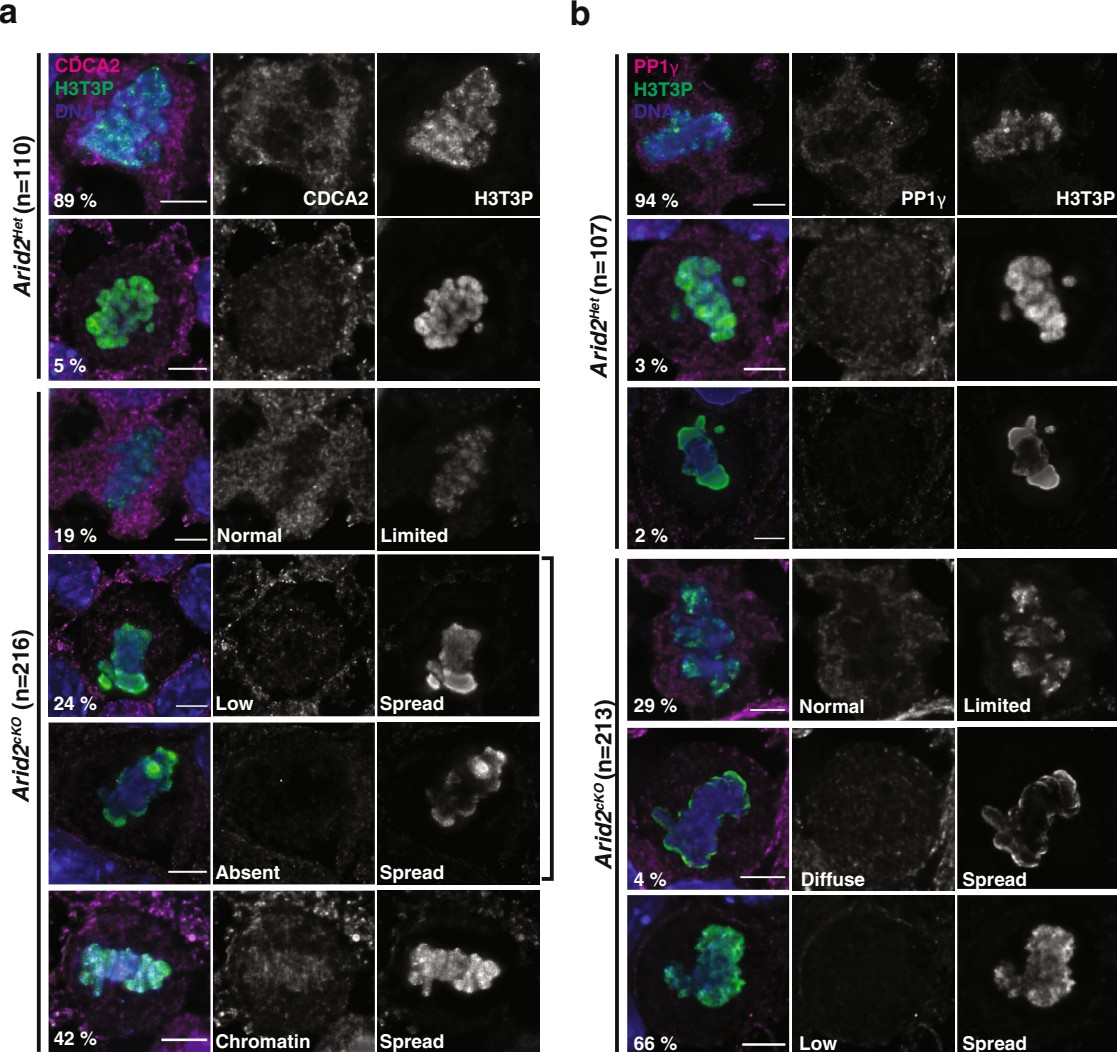

**Fig. 7 CDCA2-PP1γ abundance is regulated by ARID2.** *Arid2^Het* and *Arid2^cKO* metaphase-I spermatocytes immunolabelled for H3T3P (green) and **a** CDCA2 (magenta), **b** PP1γ (magenta). **a**, **b** DNA counterstained with DAPI (blue). Scale bars: 5 μm, magnification: 100.8x. Number (*n*) of metaphase-I spermatocytes scored and proportions (%) are indicated.

right panel), suggesting that protein stability might be affected. Therefore, *Arid2^cKO* metaphase-I spermatocytes lack a functional PP2A complex which is normally required for metaphase progression.

## Discussion

Our studies highlight a crucial requirement for ARID2 and therefore the PBAF complex in reductional male meiosis. This adds to previously identified roles of SWI/SNF in spermatogonial stem cell (SSC) maintenance and meiotic prophase-I progression[6]. It is possible that these earlier stages are regulated by the BAF complex, while PBAF activity is restricted to late meiosis-I, given that ARID2 is detected from pachynema onwards. The common theme that emerges from our current and previous studies are that SWI/SNF governs the expression of key genes to ensure successful spermatogenesis.

In this study, we show that ARID2 is essential for activities that normally promote metaphase exit such as (i) normal spindle assembly, (ii) PLK1 association with centromeres, (iii) maintaining centromere identity, and (iv) the chromosomal association of the CPC during metaphase-I (Fig. 9). Mechanistically, defects in these activities appear to be a consequence of

transcriptional mis-regulation of genes essential for cell division in the absence of ARID2. Furthermore, promoters of several cell division genes displayed PBAF occupancy evidenced by the identification of ARID2 enrichment and or peaks. While this suggests a direct role for PBAF in gene regulation, future CUT&RUN studies using *Arid2^cKO* spermatocytes will be required to confirm the PBAF genomic associations identified in this study. Phenotypically, the loss of ARID2 mimics "spindle poisons" that inhibit microtubule synthesis and activate a SAC (spindle assembly checkpoint)-induced metaphase arrest. This suggests that PBAF functions to alleviate the SAC in normal metaphase-I spermatocytes. In spermatocytes, we demonstrate that ARID2 not only activates *Plk1*, but it also interacts with PLK1, a kinetochore-associated protein. Furthermore, the centromeric association of ARID2 indicates a potential role in PLK1 recruitment. Such a mechanism might mitigate SAC in spermatocytes, given that PLK1 is a well-established regulator of anaphase onset during mitosis[63] and oocyte meiosis[21]. Other processes such as the resolution of unattached kinetochores are necessary to overcome SAC. Here, the role of ARID2 in spindle assembly is instructive. In spermatocytes, ARID2 influences centrosome maturation. Interestingly, the PBAF target, *Plk1*, was recently shown to be required for centrosome formation in mouse

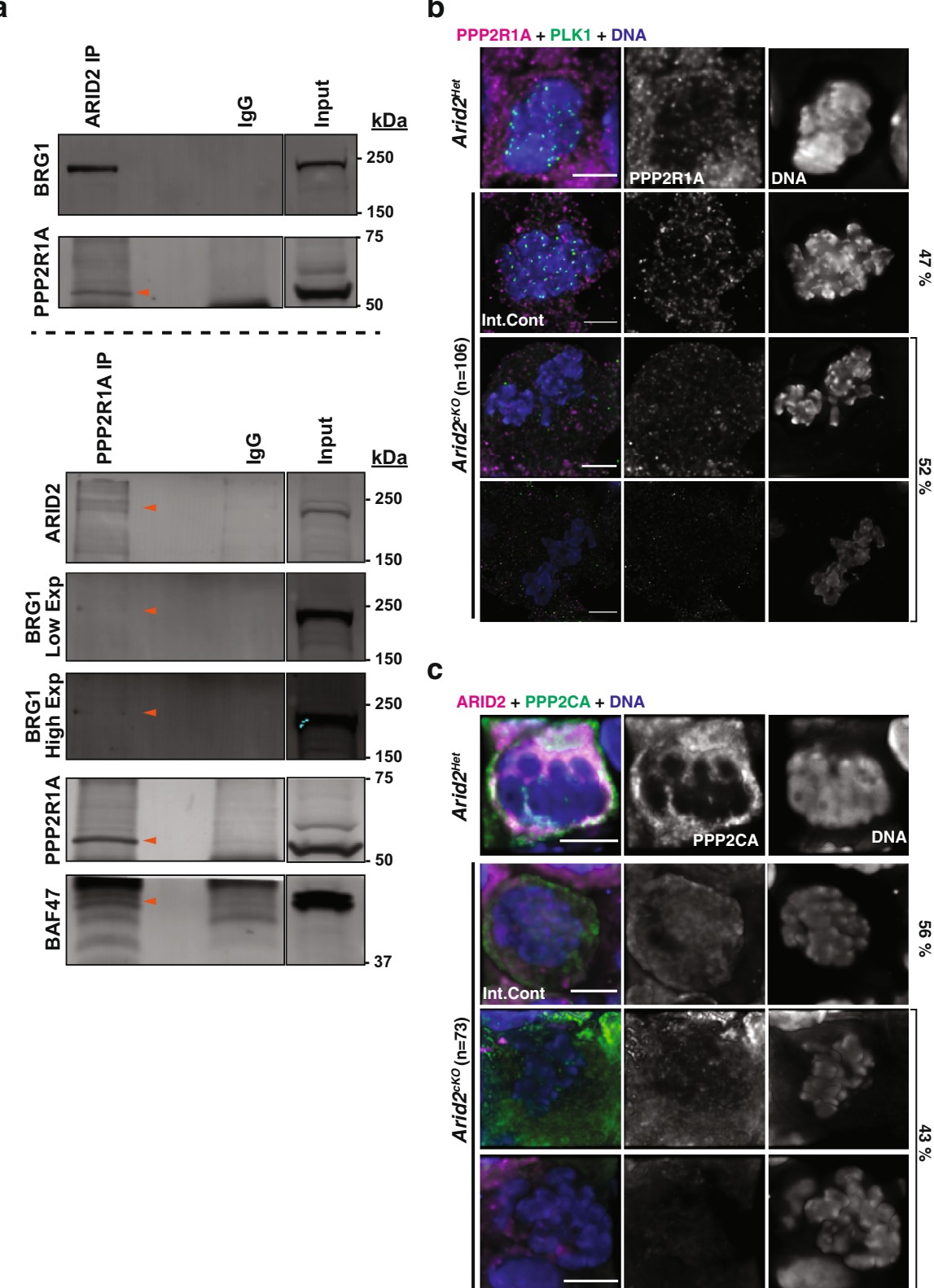

**Fig. 8 The influence of ARID2 on PP2A stability. a** ARID2 (top) and PPP2R1A (bottom) Co-IPs. Immunoblotted proteins are labeled with orange arrowheads. Data were representative of Co-IPs reproducibly repeated twice. **b, c** *Arid2*Het and *Arid2*cKO metaphase-I spermatocytes immunolabelled for **b** PPP2R1A (magenta) and PLK1 (green), **c** ARID2 (magenta) and PPP2CA (green). Scale bar: 5 µm, magnification: 100.8x. The number of metaphase-I spermatocytes scored (*n*) and their proportion (%) in *Arid2*cKO seminiferous tubules are indicated.

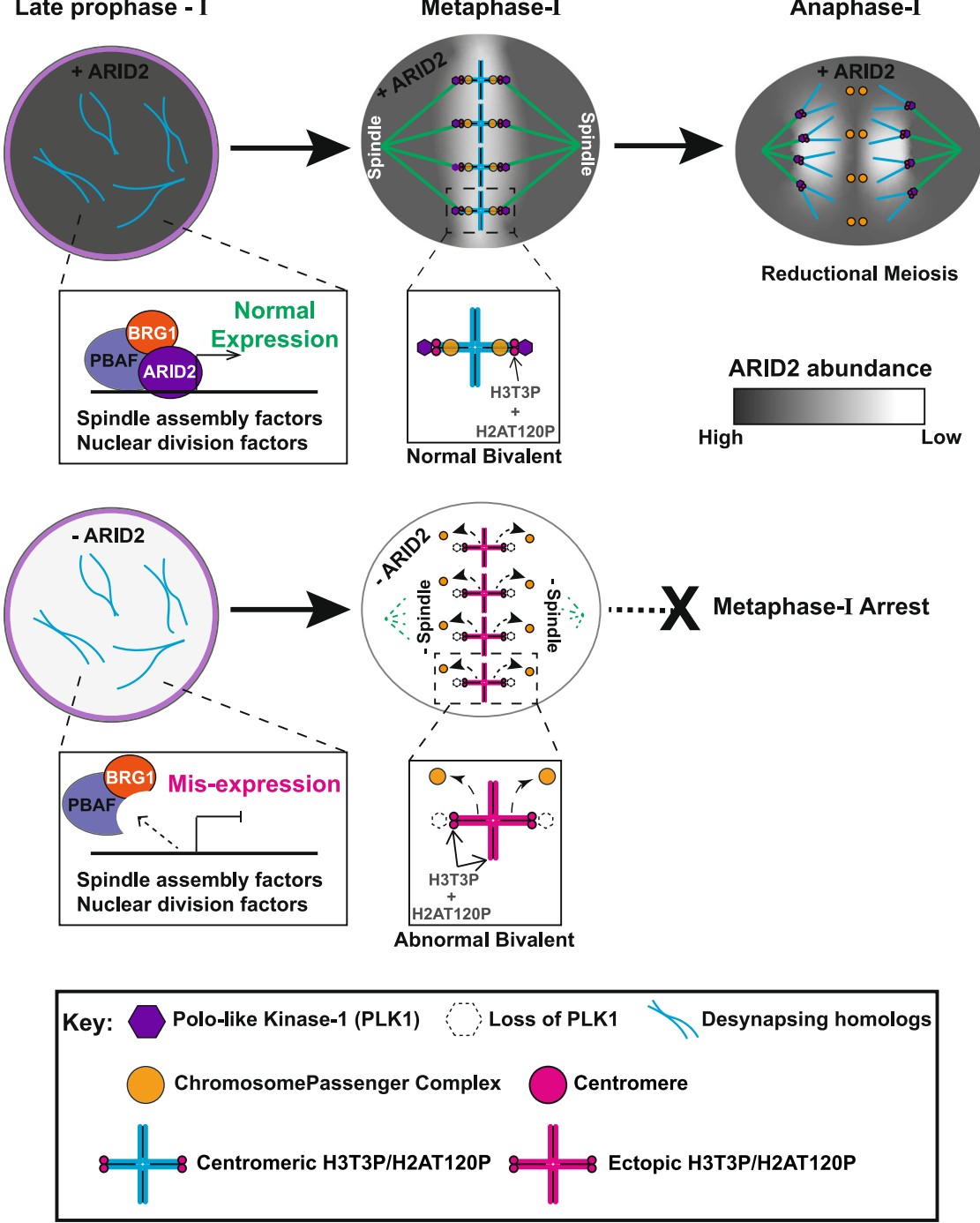

**Fig. 9 Model describing the role of PBAF in spermatocyte cell division.** ARID2 expression peaks towards the end of prophase-I (diplonema), when the PBAF complex is associated with centromeres and promoters of genes known to participate in processes such as spindle assembly and nuclear division. The timely regulation of these genes is critical for the transition from metaphase-I to anaphase-I, culminating in reductional meiosis. Metaphase-I spermatocytes expressing ARID2 assemble bipolar spindle attached to meiotic bivalents (shaded blue) that display kinetochores enriched for PLK1 (purple hexagon), centromeric H3T3P/H2AT120P (magenta circle), and chromatin-associated chromosome passenger complex (CPC) (Orange circle). The loss of ARID2 results in a metaphase-I arrest due to the mis-expression of essential cell division genes. Consequently, mutants lack normal spindle and display abnormal meiotic bivalents (shaded magenta) with kinetochores devoid of PLK1 (empty hexagon), ectopic H3T3P/H2AT120P distributed chromosomally, and chromatin deficient in CPC.

spermatocytes[45,64]. Additionally, AURKA, whose activity is influenced by ARID2 has been shown to be essential for centrosome formation in spermatocytes[45]. Of course, other mechanisms of spindle formation involving CPC[24,65], which is also subject to ARID2 dependent transcriptional regulation in spermatocytes, cannot be ruled out. While a more direct role for PBAF in meiotic spindle assembly is lacking, it is intriguing to speculate that the localization of ARID2 to spindle poles, overlapping AURKA and γ-Tubulin is physiologically relevant. Interestingly, a recent study reports that members of the PBAF complex are recruited to microtubules in mitotic cells to maintain genome stability[66].

In addition to spindle assembly, we find that ARID2 regulates the abundance of protein phosphatases during meiosis. These include PP1γ and the PP2A complex, which are essential for metaphase progression[67]. In fact, the pharmacological inhibition PP1 and PP2A is commonly used as a strategy to arrest cell at metaphase. Therefore, it is not surprising that *Arid2cKO* spermatocytes are incapable of undergoing nuclear division. Like mitotic cells, metaphase-I spermatocytes appear to rely on PP1γ and its binding partner CDCA2 to limit centromeric H3T3P, an event essential for CPC recruitment[60,68]. The novel interaction between PBAF and PP2A during meiosis is intriguing and remains a topic of future investigation. Although this association might influence PP2A stability, an alternative and exciting possibility is that PBAF recruits PP2A to regulate transcription at target loci. Evidence for a transcriptional role was recently reported in human and *Drosophila* cell lines, where PP2A regulates transcriptional pausing[69].

## Methods

**Generation of Arid2 conditional deletion and genotyping.** A knockout first allele of *Arid2* (*Arid2tm1a(EUCOMM)Wtsi*) was obtained from EUCOMM[70] and the mice were generated at the regional mutant mouse resource and research center (MMRRC; Stock number: 036982-UNC; www.mmrrc.org) at the University of North Carolina at Chapel Hill. The lacZ gene trap was "flipped out" to generate *Arid2* floxed mice (*Arid2tm1c(EUCOMM)Wtsi*) that were maintained on a mixed genetic background. Conditional knockout was achieved using *Stra8-Cre* (activated only in spermatogonia, as early as P3)[16]. *Arid2fl/fl* females were crossed to *Arid2fl+*; *Stra8-CreTg/0* males to obtain *Arid2flΔ;Stra8-CreTg/0* (*Arid2cKO*), *Arid2flΔ;Stra8-CreTg/0* (*Arid2Het*), and *Arid2fl+* (*Arid2WT*) males. Genotyping primers used in this study include *Arid2fl+* alleles - (F)- 5′-CTGCTTAGCCCAAAGGTGTC-3′; (R) 5′-GACAGTGACTTCAGCTGACC-3′, the excised allele (Δ)- forward primer used above in combination with (R) 5′-CTGAGCCCAGGTGTTTTTGT-3′ and *Stra8-Cre* - (F) 5′-GTGCAAGCTGAACAACAGGA-3′, (R) 5′-AGGGACACAGCATTG GAGTC-3′. Mice were housed under a 12 h light cycle at 20–24 °C and 30–70% humidity, at the University of North Carolina at Chapel Hill animal facility. All animal work was carried out in accordance with approved IACUC protocols at the University of North Carolina at Chapel Hill.

**Histology.** Testes and cauda epididymides from adult *Arid2WT* and *Arid2cKO* mice were dissected and fixed[3] in Bouins fixative solution (Fisher scientific Ricca chemical; 11-201) at 4 °C overnight. Tissues were dehydrated using ethanol series (50, 70, and 100%) prior to embedding in paraffin. Tissue sectioning and staining with hematoxylin and eosin, and Periodic acid-Schiff (PAS) was performed by the animal histopathology core at the University of North Carolina at Chapel Hill. Seminiferous tubule staging and cell type identification were based on staining and morphology[71,72].

**Immunofluorescence staining.** Indirect immunofluorescence (IF) was conducted on testes cryosections, squashes and spermatocyte spreads obtained from juvenile (P22–P27), and adult *Arid2WT*, *Arid2Het*, and *Arid2cKO* mice. IF for metaphase spermatocytes were performed on both testes cryosections and seminiferous tubule squashes to ensure consistent IF results across preparations.

Testis cryosections (8–10-μm thick) were prepared[6] from tissue fixed in 10% neutral buffered formalin (NBF) at 4 °C for 20 min, following which testes were sliced in half and incubated for an additional 40 min under identical fixation conditions. Fixed tissue was washed three times for 10 min each in PBS at room temperature (RT) followed by incubation in a sucrose series—10% (30 min), 20% (30 min), and 30% (1 h). Next, tissues were incubated overnight in a 1:1 solution of 30% sucrose/optimum cutting temperature (OCT, Tissue-Tek; 62550-01) formulation at 4 °C, followed by embedding in OCT, and sectioning. Antigen retrieval was performed on cryosections by boiling in 10 mM citric acid buffer (pH 6.0) for 10 min. Over this period, citrate buffer was replaced every 2 min with fresh boiling buffer and then allowed to cool for 20 min. For immunostaining, cryosections were washed in PBS at RT followed by permeabilize in 0.1% Triton X-100, blocking in antibody dilution buffer (ADB: 10% bovine serum albumin; 10% goat serum; and 0.05% Triton X-100) diluted 1:10 in PBS (ADB/PBS), for 10 min each, and incubation with primary antibody dilutions prepared in stock ADB overnight at 4 °C. The next day, samples were similarly washed in PBS, permeabilized with Triton X-100, blocked in ADB/PBS before incubating with Alexa fluor-conjugated secondary antibody dilutions prepared in ADB for 1 h at RT. Immunostained slides were washed twice in PBS/0.32% photoflo (Kodak), counterstained with DAPI, and washed once more in PBS/0.32% photoflo before mounting in Prolong Gold antifade medium (P-36931; Life Technologies).

An identical method was used to immunolabel seminiferous tubule squashes. For immunostaining involving antibodies generated from the same host, squashes

were analyzed by direct IF. Here primary antibodies were conjugated to Alexa Fluor 488/594 with Zenon rabbit IgG labeling kit (ThermoFisher; Z25370) and used for IF according to the manufacturer's protocol. Squashes were prepared using a technique described in ref. [73]. Briefly, seminiferous tubules were incubated with fixative (1% paraformaldehyde, 0.05% Triton X-100, 1X PBS) containing protease inhibitor cocktail (PIC) (Sigma: P8340) for 10 min at RT. Smaller pieces of tubules were then minced in 60 μl of fixative on a glass slide using a razor blade. After mounting with a coverslip, slides were immediately inverted onto a filter paper following which direct pressure was applied with a thumb. To ensure efficient squashing prior to coverslip removal, the excess downward pressure was generated by overlaying inverted glass slides with an Eppendorf rack bearing added weight in the form of an aluminum heating block or a filled 500 ml glass bottle for 20 min.

"3D-preserved" meiotic spreads[6,29] were prepared from spermatocyte suspensions obtained from juvenile testes. Briefly, seminiferous tubules were digested in DMEM (Dulbecco's Modified Eagle Medium) containing 1 mg/ml collagenase for 10 min on a rotator at 32 °C. Tubules were separated by centrifuging at $500 \times g$ for 3 min and then digested in DMEM containing 0.025% trypsin and 4 μg/ml DNase-I for 10 min on a rotator at 32 °C. After neutralizing trypsin with 10% fetal bovine serum (FBS), single-cell suspensions were generated by pipetting the slurry several times through a 1 ml micropipette tip and filtering it through a 70 μm filter. The resulting cells were pelleted at $500 \times g$ for 3 min, resuspended in 300–500 μl PBS and stored on ice. About 5 μl of cell suspension was added to 50 μl of 0.25% IGEPAL -CA-630 (Sigma; I8896) on a slide and incubated for 2 min at RT, followed by fixation in 100 μl of 1% paraformaldehyde (made up in 10 mM sodium borate pH 9.2) for 1 h in a humidified chamber at RT. After fixation, slides were air-dried and washed twice for 5 min each in PBS/0.32% photoflo. Immunostaining of air-dried spreads were performed using the method described for cryosections and squashes.

The entire list of primary and secondary antibodies used for IF are provided in the supplemental information (Supplementary Table 1). Images were acquired using Zeiss AxioImager-M2 and Leica Dmi8 fluorescent microscopes. Z-stacks were deconvoluted using Axiovision version 4.8.1 (AxioImager images) or Huygens essentials version 20.04 (Leica Dmi8 images) software. Image processing and fluorescence intensity measurements were performed with Fiji[74] (https://theolb.readthedocs.io/en/latest/imaging/measuring-cell-fluorescence-using-imagej.html). Briefly, regions of interest (ROI) were drawn around cells to measure the area of the cell, mean fluorescence (gray value), and integrated density. Additionally, ROI was selected in an area on the image lacking cells to measure the mean fluorescence of the background. We then calculated the corrected total cell fluorescence (CTCF) using the following equation, CTCF = integrated density of selected cell – (area of selected cell × mean fluorescence of background). Statistical significance was calculated using unpaired Student's *t*-test.

**MNase treatment of cryosections.** Frozen testis cryosections were thawed on a slide drier at 42 °C. Antigen retrieval was performed in boiling citrate buffer (pH 6.0) for 1 min, following which slides were washed twice in PBS for 5 min each. Next, testes sections were treated with 50 μl aliquots of MNase prepared at 2, 20, and 200 Kunitz units in MNase reaction buffer (New England BioLabs; M0247S) and incubated for 15 min at 37 °C. Controls (no MNase) were included and consisted of testis sections incubated with reaction buffer alone. MNase activity was stopped by adding 1/10th volume of 100 μM EDTA. Testis sections were washed in PBS before proceeding to immunofluorescence. MNase digestion was assessed by staining DNA with DAPI.

**Preparation of metaphase spreads.** DAPI-stained metaphase chromosome spreads were prepared from P19–P27 and adult *Arid2WT*, *Arid2Het*, and *Arid2cKO* testes, using a protocol described in ref. [75]. Briefly, seminiferous tubules are washed in PBS and incubated in 1% sodium citrate hypotonic solution for 12 min at RT. Following hypotonic treatment, tubules were transferred to 20 ml of fixative (three parts 95% ethanol: one part glacial acetic acid) and incubated for 15 min. Following fixation, small pieces of tubules ~1–2 inches in length were transferred to 500 μl of 60% glacial acetic acid where they become transparent due to the sloughing of cells into a suspension. After tapping the tubes to allow for homogenization, 20 μl of cell suspension was transferred to a glass slide placed on a heating block maintained at 60 °C. The cell suspension was quickly withdrawn and expelled several times on different areas of a glass slide. This was repeated twice more using fresh aliquots of cell suspension (20 μl) each time. Once dried, the slides were rehydrated in PBS for 5 min at RT, followed by DAPI staining and mounting in prolonged gold before examining chromosome spreads.

**Preparation of protein extracts.** Protein extracts were prepared from spermatogenic cells isolated from juvenile and adult testes. To obtain cells, seminiferous tubules were digested in 10 mL of DMEM (Dulbecco's Modified Eagle Medium) containing 0.5 mg/ml collagenase for 10 min on a rotator or shaking water bath at 32 °C. Digested tubules were allowed to sediment by gravity for 5 min at RT. After aspirating the supernatant, tubules were resuspended in 10 ml DMEM and allowed to settle for 5 min at RT. Sedimented tubules were transferred to 10 ml DMEM containing 0.025% trypsin and 4 μg/ml DNase-I and incubated for 10 min on a rotator or shaking water bath at 32 °C. After neutralizing trypsin with 10% fetal

bovine serum (FBS), single-cell suspensions were generated by pipetting the slurry using a plastic transfer pipette and filtering it first through a 70 µm filter and then 40 µm filter. The filtrate was centrifuged at $600 \times g$ for 5 min following which the cell pellets were washed once in PBS. To prepare nuclear extracts we used a high salt extraction method[6]. Briefly, cell pellet was resuspended in 20 PCV (packed cell volumes) of a hypotonic buffer A (10 mM HEPES-KOH pH 7.9, 1.5 mM MgCl2, 10 mM KCl, 0.1% IGEPAL, 0.5 mM DTT, 0.5 mM PMSF, and 1x PIC, 0.5) and incubated on ice for 10 min. Cells were centrifuged at $600 \times g$ for 5 min at 4 °C, resuspended in 2 PCV of buffer A and then homogenized using a dounce (type B). The homogenate was centrifuged at $930 \times g$ for 7 min at 4 °C to isolate nuclei. After washing nuclei in 10 PCV of buffer A, the pellets were resuspended in 1 ml of buffer A and centrifuged at $4820 \times g$ for 10 min at 4 °C. Resulting nuclear pellet was incubated with an equal volume of high salt buffer C (20 mM HEPES-KOH pH 7.9, 1.5 mM MgCl2, 420 mM NaCl, 10 mM KCl, 25% glycerol, 0.2 mM EDTA, 0.5 mM DTT, 0.5 mM PMSF, and 1x PIC) for 1 h on a nutator at 4 °C. The resulting nuclear lysates were diluted in 2.8x volume of buffer D (20 mM HEPES-KOH pH 7.9, 20% glycerol, 0.2 mM EDTA, 0.5 mM DTT, 0.5 mM PMSF, and 1x PIC) following which they were clarified by spinning at $13,000 \times g$ for 10 min at 4 °C, snap-frozen in liquid nitrogen and stored at −80 °C until further use. A slightly modified version of this protocol was used to prepare spermatocyte lysates for Co-immunoprecipitation (Co-IP). Here, after isolating spermatocytes, cell pellets were homogenized with a microcentrifuge tube pestle and then transferred in Buffer C containing 0.1% IGEPAL followed by incubation for 1 h at 4 °C on a nutator. The resulting lysates were diluted in 2.8x volume of buffer D. Next, they were clarified by spinning at $13,000 \times g$ for 10 min at 4 °C, and snap-frozen in liquid nitrogen and stored at −80 °C until further use.

**Western blotting**. Protein samples were separated by polyacrylamide gel electrophoresis and then transferred to PVDF (Polyvinylidene Difluoride) membranes (Bio-Rad) by wet transfer apparatus (Bio-Rad). Sample loading was assessed by staining blots with REVERT™ total protein reagents (LI-COR) or by immunoblotting for a reference protein. Blots were scanned on an LI-COR Odyssey CLx imager, viewed, and quantified using Image Studio Version 5.2.5. All antibodies used in this study and their corresponding dilutions are listed (Supplementary Table 1).

**Co-Immunoprecipitation (Co-IP)**. Co-IPs were performed using described method[6] with minor modifications. Protein extracts were thawed on ice and centrifuged at $13,000 \times g$ for 5 min at 4 °C to remove any precipitates. IP was performed with 1 mg of protein lysate of which 5% was set aside as input. Prior to IP, all lysates were treated with 25 Units of universal nuclease (Thermo scientific; 88701) for 20 min on ice. The resulting lysates were diluted in IP buffer (20 mM HEPES-KOH pH 7.9, 0.15 mM KCl, 10% glycerol, 0.2 mM EDTA, 0.5 mM DTT, 0.5 mM PMSF, and 1x PIC) to make up the volume to 1.3 ml and incubated with antibodies overnight on a rotator at 4 °C. After sequentially washing protein A (Bio-Rad: 1614013) or protein G (Invitrogen: 10003D) magnetic beads twice in PBS and once in PBS + 0.5% BSA, for 5 min each on a rotator at 4 °C, the washed beads were added to lysates and incubated at 4 °C on a rotator for 3 h to capture antibody-antigen conjugates. Following capture, beads were separated using a magnet and were washed for 5 min each on a rotator at 4 °C in the following order: twice in IP buffer, once in low salt buffer (20 mM HEPES-KOH pH 7.9, 100 mM KCl, 10% glycerol, 0.2 mM EDTA, 0.1% Tween-20, 0.5 mM DTT, 0.5 mM PMSF, and 1x PIC) and once in final wash buffer (20 mM HEPES-KOH pH 7.9, 60 mM KCl, 10% glycerol, 0.5 mM DTT, 0.5 mM PMSF, and 1x PIC). Lastly, the bead conjugated IPs were separated from wash buffer on a magnet following which proteins were eluted by resuspending the beads in 2X Laemmli buffer containing 100 mM DTT and incubating at 95 °C for 5 min.

**Preparation of acid extracted histones**. Histones were extracted from spermatogenic cells obtained from P23 and P27 $Arid2^{WT}$, $Arid2^{Het}$, and $Arid2^{cKO}$ mice by acid extraction[76]. Spermatogenic cells isolated from one testis were washed in 10 ml PBS and centrifuged at $600 \times g$ for 5 min. The cell pellet was resuspended in 1 ml hypotonic lysis buffer (10 mM Tris-Cl pH 8.0, 1 mM KCl, 1.5 mM MgCl2, and 1 mM DTT) and incubated on rotator at 4 °C for 30 min. Next, nuclei were pelleted at $10,000 \times g$ for 10 min at 4 °C. After discarding the supernatant, nuclei were resuspended in 400 µl 0.2 N HCl and incubated for 1 h at 4 °C. The sample was centrifuged at $14,000 \times g$ for 10 min at 4 °C following which supernatant containing histones were transferred to a fresh 1.5 ml tube. To precipitate histones, Trichloroacetic acid (TCA) was added to the nuclear lysates to a final concentration of 33% and incubated on ice for 30 min. Histones were pelleted by centrifugation at $14,000 \times g$ for 10 min at 4 °C. Precipitated histones were washed twice by adding ice-cold acetone without disturbing the pellet and centrifuging samples at $14,000 \times g$ for 10 min at 4 °C. After the final acetone wash, the supernatant was discarded, and pellets were air-dried at RT for 20 min followed by resuspension in 100 µl ddH2O.

**RNA extraction and quantitative RT-PCR**. Spermatocyte cell pellets isolated from P23 $Arid2^{Het}$ (n = 3) and $Arid2^{cKO}$ (n = 3) mice were subjected to RNA extraction in TRIzol reagent (Invitrogen). Total RNA was isolated and purified

using the Direct-zol RNA kit (Zymo). cDNA was synthesized using random primer mix (NEB) and ProtoScript® II reverse transcriptase (NEB; M0368L). Real-time qPCR was performed using Sso Fast EvaGreen supermix (Bio-Rad; 172-5280) on a CFX96 thermocycler (Bio-Rad). The list of primers is provided (Supplementary Table 2).

**ChIP-qPCR**. ARID2, BRG1, H3T3P, and H3K27me3 ChIP were performed in triplicates on $2 \times 10^7$ wild-type (CD-1) spermatocyte enriched populations obtained from P19 mice. Nuclei from formaldehyde-fixed cells (fixation with 0.33% formaldehyde for 30 min at 4 °C) were isolated by incubating in 1 ml Nuclei EZ lysis buffer (Sigma-NUC101) for 20 min on ice followed by centrifugation at $500 \times g$ for 5 min at 4 °C. Nuclei were resuspended in 1 ml Nuclei EZ lysis buffer and centrifuged at $930 \times g$ for 7 min. Resulting nuclear pellets were resuspended in 500 µl NUC buffer (15 mM HEPES-KOH pH 7.9, 60 mM KCl, 15 mM NaCl, 3.3 mM CaCl2, 0.32 mM Sucrose, 0.5 mM PMSF, and 1X PIC) and incubated at 37 °C for 5 min, following which 200 Kunitz units (1 µl) of Mnase (NEB; M0247) was added to the nuclei suspension and incubated for another 7.5 min at 37 °C. Mnase digestion was stopped by adding 1/50th volume of 0.5 M EGTA to samples followed by a 5-min incubation on ice. Next, 1 volume of 2X lysis buffer (20 mM Tris-Cl pH 8.0, 200 mM NaCl, 2 mM EDTA, 1 mM EGTA, 0.2% sodium deoxycholate, and 1.0% N-laurylsarcosine) and Triton X-100 to a final concentration of 1% were added to nuclei suspension, which was then passed five times each through 22 G followed by a 25 G syringe needle. Samples were incubated on ice for 1 h following which they were centrifuged at $14,000 \times g$ for 15 min at 4 °C. The supernatant containing digested chromatin was transferred to a fresh tube setting aside 10% for input. Chromatin was incubated with the antibody at 4 °C overnight on a rotator. The next day, antibody-chromatin conjugates were captured at 4 °C on a rotator for 3 h using magnetic protein A/protein G beads. Following antibody-chromatin capture, beads were separated on a magnet and washed at 4 °C on a rotator for 5 min each in the following order: twice with RIPA buffer (50 mM HEPES-KOH pH 7.5, 500 mM LiCl, 1 mM EDTA, 1% IGEPAL, and 0.7% sodium deoxycholate) and once in TE/NaCl buffer (10 mM Tris-Cl pH 8.0, 1 mM EDTA, and 50 mM NaCl). After washes, beads were separated on a magnet, resuspended in 100 µl elution buffer (1%SDS and 10 mM NaHCO3), and incubated at 65 °C on a ThermoMixer (Eppendorf) at 800 rpm for 20 min. After isolating eluate from beads using a magnetic separator, 5 µl of 5 M NaCl was added to ChIP DNA and incubated in a 65 °C water bath overnight to reverse crosslinks. DNA was digested with RNAse A for 30 min at 37 °C followed by proteinase K digestion for 1 h at 56 °C and then purified using a ChIP DNA clean and concentrator kit (Zymo) and used for qPCR. A list of primers is provided (Supplementary Table 2).

**Synchronization of spermatogenesis with WIN 18,446**. Synchronized spermatogenesis was established in mice using the potent and selective retinoic acid (RA) synthesis inhibitor, WIN 18,446 [(N,N'-1,8-Octanediylbis(2,2-dichloroacetamide)][39,77,78]. Neonatal mice were gavage-treated with WIN 18,446 (100 mM; 100 µg/g of body weight) from P1–P10; as a result, testes contained only undifferentiated type A spermatogonia. At P11, WIN 18,446 was discontinued and the mice were given a single dose of 10 µl exogenous RA (10 µg/µl) via subcutaneous injection (30-gauge needle) to initiate differentiation. Germ cells proceeded through differentiation on a predicted timeline and entered meiosis as preleptotene spermatocytes on P19. Round spermatids were observed on P29. Prophase-I (pachynema/diplonema) and metaphase spermatocyte enriched populations were isolated from P28 and P28.5 seminiferous tubules undergoing synchronous spermatogenesis using via generation of single-cell suspension.

**CUT&RUN (cleavage under targets and release using nuclease)**. Genome-wide associations of ARID2 and BRG1 in spermatogenic cells (500,000 cells/sample) obtained from P28 and P28.5 synchronized testes were determined in duplicates using the standard CUT&RUN protocol[79,80] with minor modifications. Briefly, spermatogenic cells were washed in buffer [wash buffer: 20 mM HEPES(Na) pH 7.5, 150 mM NaCl, 0.5 mM spermidine, EDTA-free protease inhibitor] three times and bound to Concanavalin A coated magnetic beads, following which bead slurry was separated on a magnet and permeabilized in antibody buffer (wash buffer + 0.05% Digitonin + 2 mM EDTA). Bead slurry was aliquoted at 50 µl into 0.2 ml 8-strip PCR tubes following which primary antibody was added to each sample and incubated on a nutator at 4 °C overnight. A no primary antibody (no IgG) control was included for P28 and P28.5 samples. The next day, bead slurry in PCR strips were separated on a magnet, washed twice in 200 µl Dig-wash buffer (wash buffer + 0.05% Digitonin), and incubated with 50 µl of secondary antibody dilution for 30 min on nutator at 4 °C. After separating bead slurry on a magnet, each sample was washed twice in 200 µl Dig-wash buffer and then incubated with 50 µl Dig-wash buffer containing 1000 ng/ml Protein A/G Micrococcal Nuclease (pA/G-MNase) on nutator at 4 °C for 1 h. Following this, samples were washed twice in 200 µl Dig-wash buffer and then resuspended in 50 µl Dig-wash buffer and allowed to chill down to 0 °C. After adding CaCl2 to a final concentration of 2 mM to each sample, tubes were incubated at 0 °C for 40 min. The MNase-digested chromatin fragments were retrieved from bead slurry by incubating in 2X STOP buffer (1000 mM NaCl, 20 mM EDTA, 4 mM EGTA, 0.05% Digitonin, and 100 µg/ml RNase A) at 37 °C for 30 min. DNA was purified using a ChIP DNA clean and concentrator kit (Zymo). Libraries were prepared using the Kapa Hyperprep kit

and sequenced on a NOVASEQ 6000 S Prime flow cell (50 bp reads, paired-end). Details of antibodies are listed (Supplementary Table 1). pA/G-MNase (Addgene ID: 123461) used for CUT&RUN was expressed and purified at the Protein Expression and Purification core at the University of North Carolina at Chapel Hill.

**CUT&RUN data analysis**. Reads were trimmed to remove Ns at either end using TrimGalore (version 0.6.2, https://www.bioinformatics.babraham.ac.uk/projects/trim_galore/) keeping the --trim-n option. Following trimming, reads were aligned to mm10 (mouse) reference genome using bowtie2[81] with the following paramters: bowtie2 --very-sensitive --no-mixed --dovetail --no-discordant --phred33 -I 10 -X 700. BAM files were generated using samtools[82]. Next, BAM files were converted to bigWig format with DeepTools, bamCoverage[83]. All bigWig files were filtered for mm10 blocklist regions[84], PCR duplicates (--ignoreDuplicates), and normalized with bamCoverage option "—scaleFactor". To calculate scaling factors, we first computed normalization factors (nf) using csaw[85] to eliminate efficiency bias between replicates. Next, we calculated the effective library size (library size X nf) which was then divided by a million to obtain a scaling factor. The reverse of the normalized scaling factor (1/normalized scaling factor per million) was supplied to "--scaleFactor" to generate normalized bigWig files. DeepTools bigWigCompare was used to generate tracks displaying $LOG_2$ ratios for each CUT&RUN sample/no IgG control. Replicate bigwig files were averaged using WiggleTools[86]. Peak calling was performed with SEACR[40]. Heatmaps were generated with DeepTools functions computeMatrix and plotHeatmap. Data were visualized on the UCSC genome browser. Gene ontology analysis was performed using Genomic Regions Enrichment of Annotation Tool (GREAT)[41] to perform.

**Reporting Summary**. Further information on research design is available in the Nature Research Reporting Summary linked to this article.

## Data availability

Data generated in this study are provided in supplementary information and source data. CUT&RUN data are deposited with GEO (Gene Expression Omnibus) under accession number GSE167539. Published RNA-seq data[13] are available from GEO under accession number GSE35005 . Published BRG1 ChIP-seq[6] data are available from GEO under accession number GSE119179. Source data are provided with this paper.

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

## Acknowledgements

We thank Magnuson lab members for their helpful comments on manuscript preparation. Dr. Jesse Raab (University of North Carolina, Chapel Hill) generated the FLPed knockout first *Arid2* floxed mice. Next-generation sequencing was performed at the Duke Center for genomic and computational biology. pA/G-MNase was produced by the Protein Expression and Purification core at the University of North Carolina at Chapel Hill (supported by Cancer Center Support Grant, P30 CA16086). We thank Dr. Michael Lampson (University of Pennsylvania), Dr. Tang K. Tang (Institute of Biomedical Sciences, Academia Sinica, Taipei), and Dr. Atilla Töth (Technische Universität Dresden) for generously providing antibodies. This work was supported by National Institutes of Health grants R01GM101974 (T.M.) and R01HD090083 (C.B.G.).

## Author contributions

D.U.M. and T.M. conceptualized and designed the project. Data curation and validation done by D.U.M. Synchronization of spermatogenesis done by O.K. and C.B.G. Writing was performed by D.U.M. reviewed and edited by C.B.G., T.M. and D.U.M. Project supervision and funding acquisition done by T.M.

## Competing interests

The authors declare no competing interests.
