## [Peer Review File · Nature Communications]

Mammalian SWI/SNF chromatin remodeler is essential for reductional meiosis in malesREVIEWER COMMENTS

Reviewer #1 (Remarks to the Author):

The manuscript by Drs. Menon and Magnuson entitled “Mammalian SWI/SNF chromatin remodeler is essential for reductional meiosis in males” reports a novel role of PBAF subunit ARID2 in the regulation of meiotic cell division. Arid2 expression peaks at pachytene and abnormality of spermatogenesis was observed in Arid2 knockout mice. Specifically, the authors reported that Arid2 conditional knockout causes aberrant spindle assembly and abnormal chromosome organization of metaphase-I spermatocytes. Mechanistically, the authors propose that Arid2 knockout cause abnormal chromosome-wide expansion of H3T3P and H2A120P, which is caused by loss of PLK1 and CDCA2 and consequently prevents chromosome organization and spindle assembly. Overall, the phenotypes as reported are robust and the conclusions are supported by results presented. Therefore, the manuscript is appropriate for Nature Communications. However, the manuscript could be improved by addressing the following concerns.

Major comments:

1. The authors propose PBAF and PLK1 interaction as a major mechanism of the observed defects. However, there is no evidence presented to support the interaction during pachytene. For example, there appears to be no co-localization between Arid2 and Plk1 (Extended data Fig. 2c and 2d). Likewise, there does not appear to be co-localization between Arid2 and PP2R1A.
2. Co-staining should be included for H3T3P and CDC2A to support the conclusion that Arid2 constrains H3T3P by regulating CDC2A.
3. The authors propose a model where the observed effects are independent of the transcriptional regulation by the Arid2 containing PBAF. Does inhibition of Plk1 or CDC2A (or aurora kinase) recapitulate the observed defects in spindle assembly and chromosome organization, respectively, during spermatogenesis?
4. It is informative to ChIP Arid2 and Brg1 to the pericentromeric region to provide a molecular link.
5. It is informative to examine changes in chromatin-associated fraction for various proteins whose distribution altered by Arid2 knockout using total protein levels as a control (Brg1, Plk1, CdcA2, Ppp2r1a, ppp2ca, Aurkc and Aurkb) in metaphase 1 spermatocytes in control and Arid2 knockout mice.
6. For the all the westernblot and IP analysis, total testis tissues or the sorted metaphase 1 spermatocytes were used? Given that Arid2 functions at the specific stage of spermatogenesis, it might be most informative to perform the analysis in the sorted cells (if feasible).
7. The quality of all the co-IP shown in the manuscript could be improved and, in particular, for Fig. 3d. In addition, reciprocal Plk1 IP and Arid2 and Brg1 westernblot should be included for Fig. 2c.

8. Given the importance of Arid2 in spermatogenesis, does Arid2 loss cause loss or impairment of male fertility?

9. For all the IF results presented (only a percentage is shown on the IF images in the present version), please provide a scatted dot plot as showed in Extended Data Fig. 2c with SD or SEM and statistical analysis.

Minor suggestions:

1. The authors previously reported that BRG1 is also essential for male meiosis. Since BRG1 level was decreased after pachynema (Fig. 1a), A discussion on the similarity and/or differences between BRG1 and ARID2 knockout in meiosis would be informative.

2. Is there any evidence in the literature that supports a correlation between Arid2 genetic alteration and male infertility? Please discuss.

3. Arid2 knockout causes accumulation of metaphase 1 spermatocytes as shown in Fig. 1d-e. However, the accumulation of pachynema did not decrease diplonema in extended data fig. 1c. Please kindly explain why this is the case.

4. Ref. 11 does not appear to support Arid2 as an early pachytene marker.

5. Please include molecular weight markers for all the westernblot and IP analysis.

Reviewer #2 (Remarks to the Author):

In this manuscript, the authors provided evidence supporting a role of Arid2, a PBAF-specific subunit, in regulating male meiosis. They showed that Stra8-Cre-dependent depletion of Arid2 in spermatogonia results in defects in spindle assembly and centromeric chromosome organization, and ultimately metaphase I arrest. Mechanistically, the authors found that depletion of Arid2 affects the association of PLK1 with centromere, and the targeting of CPC (chromosome passenger complex) to chromosome in metaphase I through regulating H2AT120 phosphorylation and the PP2A-CDCA2/PP1-H3T3P axis.

Overall, the authors provided some evidence to support their conclusions, and their findings expand our understanding of SWI/SNF function in spermatogenesis. However, some data is not very convincing. In addition, more evidence is needed to support their model. Without better quality data and more supporting evidence, the conclusions are not solid enough to justify its publication in Nature Communications.

Major points:

1. The phenotype of the Arid2 conditional knockout mice is clear. However, the underlying mechanisms are not totally clear. The model provided by the authors has several gaps:

1) It is not clear how Arid2 contributes to the recruitment of PLK1 to the centromere and the PLK1 stability?

2) Arid2 can interact with PP2A subunits (PPP2R1A and PPP2CA) to target CDCA1-PP1 and then the H3T3P to the centromere, however, without Arid2, the protein level of PP2A subunits was diminished (Fig. 3e and 3f). Its not clear how Arid2 contributes to the stability of PPP2R1A and PPP2CA. Similar explanation for the decrease of CPC complex, especially AURKB, is also lacking (Fig. 4).

2. The authors provide a hierarchical pathway in the mechanism (like PP2A-CDCA2/PP1-H3T3P-CPC-metaphase exit). To substantiate this proposed pathway, the authors should provide evidence of rescue to verify that the regulation indeed goes through such a pathway.

3. Chromatin remodelers usually functions through directly regulating chromatin organization. In this manuscript, PBAF complex regulates spermatogenesis through its interaction with PLK1 and PP2A, regulating their stability and recruitment. The authors should also look into the possibility that PBAF may regulate spermatogenesis at transcriptional level through affecting chromatin accessibility?

Minor points:

1. sFig 1a, comparing to 12wk testes, Arid2 expression pattern in P21 testes is very different and showed three or more bands. Please label the Arid2 band.

2. Most Co-IP results are relatively poor. A better quality and clearer results or quantification should be provided to make the results more conclusive.

3. Besides the MI arrest phenotype, the number of spermatocytes appear also decreased. Any explanation?

4. Fig 1d does not show accumulation of stage XII tubules. A graph might be better to show this result.

5. In Fig 3a, PLK1 expression appears abnormal and is not localized in centromere, as shown in Fig 2b and Fig 3b.

6. Fig.3b, Arid2 cko shows more PLK1 expression, especially those in cytosol, which is inconsistent with previous results (Fig 2b and Fig 3a).

Reviewer #3 (Remarks to the Author):

In this manuscript that authors seek to understand how SWI/SNF remodelers control male meiosis using a mouse model of Arid2 conditional knockout. Arid2 is deleted in spermatocytes using a Stra8-driven Cre. The authors find that although there is mosaicism in the Cre function to delete Arid2, spermatocytes that do delete the gene are defective. Consistent with defects, KO testes are smaller, and many spermatocytes fail to complete meiosis I. The authors then investigate the cause of this MI arrest and use immunofluorescence to probe chromosome segregation regulators such as Plk1, Aurora kinases, PP1 phosphatase subunits, and histone marks (H3S10ph, H3T3ph and H2AT120ph) and immunoprecipitation assays. What is convincing is that centromeric Plk1 is lost in the KO, explaining spindle abnormalities and MI arrest. By the images presented here, I am not as convinced by the conclusions that Arid2 is required for centromere ID and targeting the CPC to chromosomes. I explain these concerns and others in detail below.

1. In general, the use of epifluorescence based imaging for many of the antibodies used here is not providing the resolution that is essential for accurate analysis and intensity measurements. I have a major concern that there are data misinterpretations because of this method and would be more convince if confocal based microscopy was used.

2. Methods:

a. In addition to the expectation of confocal microscopy that readers will expect to see, there is no detail about how images and intensities were measured. Were centromeric regions of interest isolated, background subtracted, or were entire areas taken? How was “spreading” vs “low” quantified?

b. For westerns, images of the entire length of the lane are typically required.

c. Did the authors do fertility trials with these animals? Based on the Cauda epididymis image, I would expect infertility, but it would be useful to confirm this.

3. Figure 1:

a. Should panel A be presented as a bar graph at each developmental time point? That is, I don't think the same cells were analyzed as a time continuum.

b. Panel B- the abbreviation “rSt” is not helpful when the rest of the labels are cell cycle stages. What stage of meiosis are “rSt” in?

c. Panel e is not useful for the scope of this manuscript. What would be more useful is a quantification of how many spermatocytes are arrested at Metaphase I.

4. Figure S1-

a. I recommend using an arrow to indicate the Arid2 band in the western for the young animal testes. Can you quantify the level of KO? This would give the readers the sense of the % of spermatocytes that Cre did not work in.

b. Panel D- I am not convinced with the ARID2/BRG1 IF. I only see background staining, nothing specific to indicate spindle midzone localization. This is also beyond the scope of this MS so I would recommend to remove this panel.

5. Line 64 describes Arid2 localization as cytosolic with a “narrow localization to chromosomes.” I agree most looks cytosolic. I do not see an enrichment near chromosomes as the authors indicate. Using this point to set up rationale of looking at chromosome segregation regulators is therefore not strong.

6. Figure 2:

a. Panel a: What % of spindle + are in controls? If H3S10ph is normal, how can the authors explain the defect in Aurora kinase localization?

b. Panel b: What % of spermatocytes are Plk1+ in controls? The authors conclude in line 197 that Arid2 is required to “maintain centromere identity” but CenpA, the epigenetic centromere mark, is normal in KO. Therefore, I think this conclusion is not supported.

c. Panel c: What is this “non-specific streak” in the Arid2 IP blots? These are empty lanes based on the lower blot image, so why are there bands here?

7. Figure S2:

a. Panel c: The authors need to explain why some KO spermatocytes don't have plk1 signal but do have Arid2 (right-most lane of scatter plot graph). This highlights the importance to include WT quantifications throughout the MS (which is largely missing).

8. Figure 3:

a. Panel a: H3T3ph “spreading” vs low” are opposite results and therefore inconsistent with one another. This therefore makes me think that the imaging approach and analysis are problematic.

b. Panel b: I don't agree that H2AT120p is “spread” in this image. This looks like antibody stickiness around the perimeter of the chromosomes. Is this representative? If it is, then the classification is not accurate. What do the other 38% of KO look like?

c. Panel e: The authors describe the localization of PPP2R1A as “on chromatin and cytosolically.” I only see cytosol localization here.

9. Figure 4:

a. Panel b: The AURKC IF looks non-specific. How was this quantified?

b. Panel c: The AURKB IF is convincing. However, the result is not consistent with “spreading” of H3T3p and H2AT120ph. If these marks are spread, then they will recruit more CPC. This was shown in oocytes and tissue culture cells. In oocytes, when the H3T3 kinase, Haspin, was overexpressed, there was increased H3T3p and increased AURKC-CPC on chromosomes. Line 228 is not reporting the literature accurately. Therefore, this result brings me back to my primary concern about image acquisition and analysis leading to mis-interpretation.

Authors response

We appreciate the comments provided by the reviewers and have made major revisions to address their concerns. These revisions have significantly improved the quality of the data and reveal greater insight into the mechanism underlying the role of PBAF complex during male meiosis. Briefly, we show that the PBAF complex associates with centromeres and promoters of genes known to effect processes such as spindle assembly and nuclear division. The timely regulation of these genes is critical for the transition from metaphase-I to anaphase-I, culminating in reductional meiosis. We also demonstrate a requirement for PBAF in maintaining chromatin accessibility during metaphase-I. We have deposited raw and processed genomic data with GEO (Accession no: **GSE167539**). To access the data reviewers will need to provide the secure token number: **cjmbgqselhedrkd**

Below you will find our response to each reviewer comment/question.

REVIEWER COMMENTS

Reviewer #1 (Remarks to the Author):

The manuscript by Drs. Menon and Magnuson entitled “Mammalian SWI/SNF chromatin remodeler is essential for reductional meiosis in males” reports a novel role of PBAF subunit ARID2 in the regulation of meiotic cell division. Arid2 expression peaks at pachytene and abnormality of spermatogenesis was observed in Arid2 knockout mice. Specifically, the authors reported that Arid2 conditional knockout causes aberrant spindle assembly and abnormal chromosome organization of metaphase-I spermatocytes. Mechanistically, the authors propose that Arid2 knockout cause abnormal chromosome-wide expansion of H3T3P and H2A120P, which is caused by loss of PLK1 and CDCA2 and consequently prevents chromosome organization and spindle assembly. Overall, the phenotypes as reported are robust and the conclusions are supported by results presented. Therefore, the manuscript is appropriate for Nature Communications. However, the manuscript could be improved by addressing the following concerns.

Major comments:

1. The authors propose PBAF and PLK1 interaction as a major mechanism of the observed defects. However, there is no evidence presented to support the interaction during pachytene. For example, there appears to be no co-localization between Arid2 and Plk1 (Extended data Fig. 2c and 2d). Likewise, there does not appear to be co-localization between Arid2 and PP2R1A.

In the revised manuscript we show that ARID2 regulates the expression of *Plk1*. Therefore, PBAF influences PLK1 association with kinetochores by regulating its abundance.

We disagree with the comment that our data fail to support an interaction with PLK1/PPP2R1A. Using a standard technique such as Co-IP we demonstrate an unambiguous association between PLK1 and PBAF (ARID2/BRG1) (Fig. 2c). Furthermore, using a high resolution technique such as ChIP, we show that that ARID2 and BRG1 associate with centromeres in spermatogenic cells enriched for pachytene/diplotene spermatocytes (Fig. 4c). These data clearly indicate that PBAF localizes proximal to PLK1 during meiosis. It should be noted that during pachynema/diplonema ARID2 is diffusely spread through-out the nucleus. Therefore, its association is not restricted to PLK1 sites as would be expected for a chromatin factor.

In the case of the PP2A complex, we confirm physical associations between PBAF and PPP2R1A by Co-IP (Fig 8a). By IF, the scaffolding subunit PPP2R1A, overlaps ARID2 in the cytosol of metaphase-I spermatocytes (compare fig. 8b to fig. 8c/Extended fig.1d). Additionally, PP2A catalytic subunit, PPP2CA clearly overlaps ARID2 in the cytosol during metaphase-I (Fig. 8c).

2. Co-staining should be included for H3T3P and CDC2A to support the conclusion that Arid2 constrains H3T3P by regulating CDC2A.

This has been done. While CDCA2 is perturbed in the presence of ectopic H3T3P (Fig. 7a), we found that the major reason for the failure to limit H3T3P was the deficiency in CDCA2 interacting partner, PP1 γ , upon the loss of ARID2 (Fig. 7b)

3. The authors propose a model where the observed effects are independent of the transcriptional regulation by the Arid2 containing PBAF. Does inhibition of Plk1 or CDC2A (or

aurora kinase) recapitulate the observed defects in spindle assembly and chromosome organization, respectively, during spermatogenesis?

In the revised manuscript using CUT&RUN we mapped ARID2 and BRG1 at promoters of genes essential for cell division. These include *Cdca2* and *Pp1γ* (Extended fig 6a) that are transcriptionally repressed upon the loss of ARID2 (Extended fig 6b). While we do not detect strong ARID2/BRG1 enrichment at *Plk1* promoters its mRNA abundance is significantly down-regulated upon the loss of ARID2 (Extended fig. 2d).

4. It is informative to ChIP Arid2 and Brg1 to the pericentromeric region to provide a molecular link.

We have done this and present evidence to show that ARID2 and BRG1 associate with centromeric minisatellites while being weakly enriched at pericentric major satellites (Fig. 4c).

5. It is informative to examine changes in chromatin-associated fraction for various proteins whose distribution altered by Arid2 knockout using total protein levels as a control (Brg1, Plk1, CdcA2, Ppp2r1a, ppp2ca, Aurkc and Aurkb) in metaphase 1 spermatocytes in control and Arid2 knockout mice.

For PLK1, CDCA2, PP2A and AURKB/AURKC, the IF analyses illustrate the changes in their abundance/subcellular localization in individual metaphase-I spermatocytes. In the case of BRG1, its abundance appears unperturbed upon ARID2 loss (data not shown). The inability to separate sufficient numbers of metaphase-I spermatocytes from pachytene/diplotene would confound the interpretation of results using chromatin fractionation technique suggested.

6. For the all the western blot and IP analysis, total testis tissues or the sorted metaphase 1 spermatocytes were used? Given that Arid2 functions at the specific stage of spermatogenesis, it might be most informative to perform the analysis in the sorted cells (if feasible).

In normal testis metaphase-I spermatocytes represent ~ 4 % of total spermatocytes. For our IP and WB analyses we used mixed populations enriched for pachytene/diplotene spermatocytes along with fewer metaphase-I spermatocytes. Sorting metaphase-I spermatocytes is technically challenging given that they share similar size and DNA content with late prophase-I cells.

7. The quality of all the co-IP shown in the manuscript could be improved and, in particular, for Fig. 3d. In addition, reciprocal Plk1 IP and Arid2 and Brg1 westernblot should be included for Fig. 2c.

These have been addressed in the revised manuscript.

8. Given the importance of Arid2 in spermatogenesis, does Arid2 loss cause loss or impairment of male fertility?

The absence of spermatids in the testes combined with the absence of sperm in cauda epididymis indicate that males are infertile.

9. For all the IF results presented (only a percentage is shown on the IF images in the present version), please provide a scatted dot plot as showed in Extended Data Fig. 2c with SD or SEM and statistical analysis.

We have provided dot plots for quantification of ARID2, PLK1, AURKA, PP1 γ and H3T3P fluorescence with corresponding SEM in the revised manuscript. Details of statistical analyses are provided. For phenotypes such as the absence/presence of spindle and the broad/limited distribution of H3T3P/H2AT120P that are qualitative in nature, we have indicated the proportion of cells displaying defects as has been done in other published studies.

Minor suggestions:

1. The authors previously reported that BRG1 is also essential for male meiosis. Since BRG1 BRG1 level was decreased after pachynema (Fig. 1a), A discussion on the similarity and/or differences between BRG1 and ARID2 knockout in meiosis would be informative.

A brief discussion of the similarity and difference are included in the introduction and discussion.

2. Is there any evidence in the literature that supports a correlation between Arid2 genetic alteration and male infertility? Please discuss.

There is no reported evidence indicating that ARID2 impact mammalian fertility to the best of our knowledge

3. Arid2 knockout causes accumulation of metaphase 1 spermatocytes as shown in Fig. 1d-e.

However, the accumulation of pachynema did not decrease diplonema in extended data fig. 1c. Please kindly explain why this is the case.

We are not exactly sure that we understand the question. The proportion of diplonema are slightly reduced in *Arid2*^{CKO} relative to the control. From our study ARID2 appears to be dispensable for prophase-I of meiosis. Meiotic DNA double strand break repair and chromosomal pairing appear normal in the absence of ARID2 (data not shown).

4. Ref. 11 does not appear to support Arid2 as an early pachytene marker.

The data is presented in the supplementary data 2 associated with the cited article (Ernst et al, PMID: 30890697).

5. Please include molecular weight markers for all the western blot and IP analysis.

The molecular weight markers have been incorporated into all western blots.

Reviewer #2 (Remarks to the Author):

In this manuscript, the authors provided evidence supporting a role of Arid2, a PBAF-specific subunit, in regulating male meiosis. They showed that Stra8-Cre-dependent depletion of Arid2 in spermatogonia results in defects in spindle assembly and centromeric chromosome organization, and ultimately metaphase I arrest. Mechanistically, the authors found that depletion of Arid2 affects the association of PLK1 with centromere, and the targeting of CPC (chromosome passenger complex) to chromosome in metaphase I through regulating H2AT120 phosphorylation and the PP2A-CDCA2/PP1-H3T3P axis.

Overall, the authors provided some evidence to support their conclusions, and their findings expand our understanding of SWI/SNF function in spermatogenesis. However, some data is not very convincing. In addition, more evidence is needed to support their model. Without better quality data and more supporting evidence, the conclusions are not solid enough to justify its publication in Nature Communications.

Major points:

1. The phenotype of the Arid2 conditional knockout mice is clear. However, the underlying mechanisms are not totally clear. The model provided by the authors has several gaps:
 - 1) It is not clear how Arid2 contributes to the recruitment of PLK1 to the centromere and the PLK1 stability?

In the revised manuscript we show that *Plk1* is activated by ARID2 in spermatocytes. The loss of ARID2 results in reduced mRNA abundance in late prophase-I spermatocytes which contributes to protein depletion in metaphase-I (Extended fig. 2c,d).

- 2) Arid2 can interact with PP2A subunits (PPP2R1A and PPP2CA) to target CDCA1-PP1 and then the H3T3P to the centromere, however, without Arid2, the protein level of PP2A subunits

was diminished (Fig. 3e and 3f). Its not clear how Arid2 contributes to the stability of PPP2R1A and PPP2CA. Similar explanation for the decrease of CPC complex, especially AURKB, is also lacking (Fig. 4).

Our CUT&RUN data show that *Ppp2r1a* and *Ppp2ca* promoters are targeted by ARID2. However, the transcription of these genes remains unperturbed upon the loss of ARID2. Therefore, we conclude that the deficiency in PP2A might result from poor protein stability in the absence of ARID2.

In the case of the CPC kinases, both *Aurkc* and *Aurkb* were identified as ARID2 target genes (Extended fig.6a). The loss of ARID2 significantly reduces *Aurkc* transcript abundance thereby resulting in a deficiency in chromatin bound AURKC. *Aurkb* mRNA abundance is also reduced albeit to a lesser extent when compared to *Aurkc* upon ARID2 loss (Extended fig. 6b). We reexamined AURKB localization by IF to find that most mutants displayed a diffuse staining pattern without the characteristic centromeric association seen in control metaphase-I spermatocytes. Very few metaphase-I spermatocytes displayed the complete loss of AURKB upon the loss of ARID2 (Fig. 6c,d). The defect in AURKB mis localization might result from transcriptional mis regulation combined with a loss in AURKC.

2. The authors provide a hierarchical pathway in the mechanism (like PP2A-CDCA2/PP1-H3T3P-CPC-metaphase exit). To substantiate this proposed pathway, the authors should provide evidence of rescue to verify that the regulation indeed goes through such a pathway. We acknowledge that our proposed model presents certain gaps. While rescue experiments are definitely worth pursuing, they will be time intensive given that we would have to generate several of the required transgenic mouse lines. As such these experiments while interesting represent potential topics of future investigation.

3. Chromatin remodelers usually functions through directly regulating chromatin organization. In this manuscript, PBAF complex regulates spermatogenesis through its interaction with PLK1 and PP2A, regulating their stability and recruitment. The authors should also look into the possibility that PBAF may regulate spermatogenesis at transcriptional level through affecting chromatin accessibility?

We performed CUT&RUN and ChIP to show that ARID2 and BRG1 occupy the promoters of effectors of cell division and associates with centromeres during meiosis. The transcriptional mis-regulation of many target genes underlie the mutant phenotypes characterized in this study.

We also performed a MNase assay to show that PBAF is essential to maintain chromatin accessibility during metaphase-I.

Minor points:

1. sFig 1a, comparing to 12wk testes, Arid2 expression pattern in P21 testes is very different and showed three or more bands. Please label the Arid2 band.

Changes have been made.

2. Most Co-IP results are relatively poor. A better quality and clearer results or quantification should be provided to make the results more conclusive.

We repeated Co-IP's to generated better quality blots.

3. Besides the MI arrest phenotype, the number of spermatocytes appear also decreased. Any explanation?

Our data does not indicate a decrease in number of prophase-I spermatocytes (Fig. 1e, Ext. fig 1c).

4. Fig 1d does not show accumulation of stage XII tubules. A graph might be better to show this result.

A graph indicating the proportion of stage XII tubules observed in control and *Arid2^{cko}* testes has been added to Fig1.d

5. In Fig 3a, PLK1 expression appears abnormal and is not localized in centromere, as shown in Fig 2b and Fig 3b.

This point is not clear to us. PLK1 displays the expected localization at centromeres. PLK1 foci are clearly visible at the outward facing tips of metaphase-I chromosomes in WT and internal controls.

6. Fig.3b, Arid2 cko shows more PLK1 expression, especially those in cytosol, which is inconsistent with previous results (Fig 2b and Fig 3a).

This does not take away from the most obvious defect which is that centromeric PLK1 is lost in the absence of ARID2. Additional quantification data in Extended fig. 2c show that mutant

metaphase-I spermatocytes are deficient in PLK1 upon the loss of ARID2. We have replaced the image for mutant metaphase-I spermatocytes in Fig.3b with one that is more representative. Additionally, we have also added an image representing internal control.

Reviewer #3 (Remarks to the Author):

In this manuscript that authors seek to understand how SWI/SNF remodelers control male meiosis using a mouse model of Arid2 conditional knockout. Arid2 is deleted in spermatocytes using a Stra8-driven Cre. The authors find that although there is mosaicism in the Cre function to delete Arid2, spermatocytes that do delete the gene are defective. Consistent with defects, KO testes are smaller, and many spermatocytes fail to complete meiosis I. The authors then investigate the cause of this MI arrest and use immunofluorescence to probe chromosome segregation regulators such as Plk1, Aurora kinases, PP1 phosphatase subunits, and histone marks (H3S10ph, H3T3ph and H2AT120ph) and immunoprecipitation assays. What is convincing is that centromeric Plk1 is lost in the KO, explaining spindle abnormalities and MI arrest. By the images presented here, I am not as convinced by the conclusions that Arid2 is required for centromere ID and targeting the CPC to chromosomes. I explain these concerns and others in detail below.

1. In general, the use of epifluorescence based imaging for many of the antibodies used here is not providing the resolution that is essential for accurate analysis and intensity measurements. I have a major concern that there are data misinterpretations because of this method and would be more convince if confocal based microscopy was used.

We disagree with reviewer #3's characterization of our data as being misinterpreted. Many of the phenotypes we report such as the loss of spindle, centromeric loss of PLK1 and ectopic distribution of H3T3P/H2AT120P are striking. It is unclear how confocal microscopy would change the overall conclusions of these studies. Our IF results are consistent with what has been cited in literature related to meiosis and mitosis. Also, the mutant phenotypes are consistent with a defect in spermatocyte division and agree with the genomic and qRT-PCR data we have added to the revised manuscript.

2. Methods:

a. In addition to the expectation of confocal microscopy that readers will expect to see, there is no detail about how images and intensities were measured. Were centromeric regions of interest isolated, background subtracted, or were entire areas taken? How was "spreading" vs "low" quantified?

The method for Intensity measurement has been provided in the manuscript. We measure the corrected total cell fluorescence, which is calculated by subtracting the mean background fluorescence for the area selected from the integrated density for selected area (metaphase-I spermatocyte). In the case of H3T3P and H2AT120P, 'spreading' and 'low' identifiers are based on the distribution of these histone marks in the population of *Arid2^{CKO}* spermatocytes scored. We have provided the proportions associated with each phenotype reported in the manuscript.

b. For westerns, images of the entire length of the lane are typically required.

In the case of Co-IP's each membrane was cut to allow for the development of multiple protein bands falling within different size ranges. In the revised manuscript the western blots with molecular weight markers are provided. For the Histone WBs, entire lanes have been shown along with protein loading for each lane. The membrane itself was cut to the size range of 25kDa-10kDa before WB. There is no point of using the entire membrane (> 25kDa) for histone WB.

c. Did the authors do fertility trials with these animals? Based on the Cauda epididymis image, I would expect infertility, but it would be useful to confirm this.

The reviewer is right about expecting infertility. As stated above the absence of sperm in epididymis obviates the need to test fertility. Even *Arid2^{CKO}* testes are devoid of mature spermatids.

3. Figure 1:

a. Should panel A be presented as a bar graph at each developmental time point? That is, I don't think the same cells were analyzed as a time continuum.

These were bulk-RNA seq data obtained from purified germ cell populations. Since spermatogenesis is a continuous process, we think it is helpful to represent the data as a line graph to represent the transcript abundance of related SWI/SNF subunits. This allows for the visualization of multiple subunits across germ cell populations presented on the X-axis in order of their appearance.

b. Panel B- the abbreviation "rSt" is not helpful when the rest of the labels are cell cycle stages. What stage of meiosis are "rSt" in?

“rSt” denotes round spermatid. This has been changed to the more typical abbreviation ‘RT’.

c. Panel e is not useful for the scope of this manuscript. What would be more useful is a quantification of how many spermatocytes are arrested at Metaphase I.

We have added a graph to panel e indicating the proportion of stage XII tubules in control and *Arid2^{ckO}* testes.

4. Figure S1-

a. I recommend using an arrow to indicate the Arid2 band in the western for the young animal testes. Can you quantify the level of KO? This would give the readers the sense of the % of spermatocytes that Cre did not work in.

The WB has been annotated with arrows indicating ARID2 band in juvenile animal. A better sense of Cre efficiency can be derived from the number of internal controls scored in each IF experiments. We have provided the percentage of internal controls.

b. Panel D- I am not convinced with the ARID2/BRG1 IF. I only see background staining, nothing specific to indicate spindle midzone localization. This is also beyond the scope of this MS so I would recommend to remove this panel.

The loss of signal in *Arid2^{ckO}* metaphase-I spermatocytes (Ext Fig 2) argues against the spindle associated signal being “background”. In Anaphase-I, ARID2 and BRG1 signal can be detected from the area between the segregating chromosome masses (spindle midzone). Furthermore previous studies (Hodges et al., 2018) and a recent preprint (Karki et al., 2020) indicate that SWI/SNF is associated with spindle during mitosis.

5. Line 64 describes Arid2 localization as cytosolic with a “narrow localization to chromosomes.” I agree most looks cytosolic. I do not see an enrichment near chromosomes as the authors indicate. Using this point to set up rationale of looking at chromosome segregation regulators is therefore not strong.

Metaphase-I arrest associated with the loss of ARID2 indicates that cell division might be affected. In the revised manuscript we show that ARID2 occupies promoters of many genes

essential for cell division. Additionally, we identify ARID2 and BRG1 enrichment at centromeric satellite repeats by CHIP-qPCR. Therefore it is reasonable to investigate the role of ARID2 in chromosome segregation.

6. Figure 2:

a. Panel a: What % of spindle + are in controls? If H3S10ph is normal, how can the authors explain the defect in Aurora kinase localization?

All WT controls that we scored had spindle. The IF was extremely efficient for spindle. More so over mutants were compared to internal controls which makes our interpretations reliable. The proportions of internal controls are indicated in the figures. Phosphorylation of H3S10P occurs prior to metaphase-I in spermatocytes. H3S10P is catalyzed normally in *Arid2*^{CKO} pachytene and Diplotene spermatocytes (data not shown). Furthermore studies from the Schindler lab have shown that the loss of both AURKB and AURKC do not impact H3S10P during oocyte meiosis (Nguyen et al., 2018). Therefore, the establishment of H3S10P is unaffected by the loss of ARID2 during meiosis.

b. Panel b: What % of spermatocytes are Plk1+ in controls? The authors conclude in line 197 that Arid2 is required to “maintain centromere identity” but CenpA, the epigenetic centromere mark, is normal in KO. Therefore, I think this conclusion is not supported.

All WT controls had PLK1⁺. Internal controls were critical in accurately interpreting the mutant phenotype. The numbers on internal controls are provided in the manuscript. Internal controls are directly comparable to mutants as they are obtained from the same sample (squash/section) and treated identically to the mutants with respect to IF conditions. In the manuscript we state that ARID2 is required to “maintain centromere identity” given its effects on cell division associated centromeric markers such as H3T3P and H2AT120P.

c. Panel c: What is this “non-specific streak” in the Arid2 IP blots? These are empty lanes based on the lower blot image, so why are there bands here?

Better quality Co-IP's have been included in the revised manuscript.

7. Figure S2:

a. Panel c: The authors need to explain why some KO spermatocytes don't have plk1 signal but

do have Arid2 (right-most lane of scatter plot graph). This highlights the importance to include WT quantifications throughout the MS (which is largely missing).

There was an error in classification of PLK1⁻ cells. Furthermore our method of quantification did not correct the total cell fluorescence for background signal. We have re-quantified all the data in the revised manuscript. Furthermore, in all figures we have provided comparisons to internal controls, which makes the interpretation of our IF data reliable.

8. Figure 3:

a. Panel a: H3T3p “spreading” vs low” are opposite results and therefore inconsistent with one another. This therefore makes me think that the imaging approach and analysis are problematic.

As per reviewer #1’s comments we co-stained for CDCA2 and H3T3P. From this study it was clear that a proportion of mutant metaphase-I spermatocytes retain CDCA2 association with chromatin (Fig. 7a). We further go onto show that the presence of chromosome-wide H3T3P is associated with a deficiency in its cognate phosphatase, PP1 γ . However certain mutant cells still exhibit PP1 γ expression that might be sufficient to target H3T3P (Fig. 7 b and Extended fig. 7c). We have reported all the localization patterns and the proportions in which they occur. Therefore, we don’t not believe there are problems with our imaging.

b. Panel b: I don’t agree that H2AT120p is “spread” in this image. This looks like antibody stickiness around the perimeter of the chromosomes. Is this representative? If it is, then the classification is not accurate. What do the other 38% of KO look like?

If there was antibody stickiness, the controls would display non-specific staining. Furthermore, internal controls from *Arid2*^{CKO} squashes do not display any stickiness. The remaining 38% constitute internal controls. Internal control has been added to Fig. 3b. We agree that H2AT120P delocalization from centromeres is not as severe as seen with H3T3P. We have modified the description of the phenotype to reflect this point.

c. Panel e: The authors describe the localization of PPP2R1A as “on chromatin and cytosolically.” I only see cytosol localization here.

While most of the signal appears to be distributed in the cytoplasm, the chromatin associated signal is observed at lower abundance in control metaphase-I spermatocytes.

9. Figure 4:

a. Panel b: The AURKC IF looks non-specific. How was this quantified?

The phenotype is qualitative and compared to controls and internal controls. If AURKC staining was non-specific we would not expect the mutants to display diminished chromatin association of AURKC. Additionally, we show that this defect is associated with reduced *Aurkc* mRNA abundance and consistent with a loss of AURKC, we show that its substrate pINCENP is also reduced upon ARID2 loss.

b. Panel c: The AURKB IF is convincing. However, the result is not consistent with “spreading” of H3T3ph and H2AT120ph. If these marks are spread, then they will recruit more CPC. This was shown in oocytes and tissue culture cells. In oocytes, when the H3T3 kinase, Haspin, was overexpressed, there was increased H3T3p and increased AURKC-CPC on chromosomes. Line 228 is not reporting the literature accurately. Therefore, this result brings me back to my primary concern about image acquisition and analysis leading to mis-interpretation.

We repeated the AURKB IF and include additional data obtained by co-staining for AURKB and H3T3P. Majority of *Arid2^{ckO}* metaphase-I spermatocytes display a diffuse AURKB staining pattern while the remaining lack AURKB in the presence of chromosome-wide H3T3P. Additionally, the ARID2 and BRG1 CUT&RUN and qRT-PCR data show that AURKB and AURKC, the CPC kinases are transcriptionally regulated by ARID2. Therefore, deficiencies in AURKC/AURKB underlie the loss of CPC despite genome-wide H3T3P distribution.

References:

- Hodges, H. C., Stanton, B. Z., Cermakova, K., Chang, C. Y., Miller, E. L., Kirkland, J. G., Ku, W. L., Veverka, V., Zhao, K. and Crabtree, G. R.** (2018). Dominant-negative SMARCA4 mutants alter the accessibility landscape of tissue-unrestricted enhancers. *Nat. Struct. Mol. Biol.* **25**, 61–72.
- Karki, M., Jangid, R. K., Seervai, R. N. H., Bertocchio, J.-P., Hotta, T., Msaouel, P., Jung, S. Y., Grimm, S. L., Coarfa, C., Weissman, B. E., et al.** (2020). A Cytoskeletal Function for PBRM1 Reading Methylated Microtubules. *bioRxiv* 2020.04.21.053942.
- Nguyen, A. L., Drutovic, D., Vazquez, B. N., El Yakoubi, W., Gentilello, A. S., Malumbres, M., Solc, P. and Schindler, K.** (2018). Genetic Interactions between the Aurora Kinases Reveal New Requirements for AURKB and AURKC during Oocyte Meiosis. *Curr. Biol.* **28**, 3458-3468.e5.

REVIEWER COMMENTS

Reviewer #1 (Remarks to the Author):

This reviewer thanks the authors for addressing my comments experimentally or by explanation/clarification. This is a nice story that expands the functional role of ARID2-containing PBAF complex in meiotic division. Congratulations. Rugang Zhang

Reviewer #2 (Remarks to the Author):

This is a revised manuscript. In the previous submission, I raised three major concerns regarding the potential mechanism leading to the observed phenotype (major comment #1), requested a rescue experiment to support their proposed hierarchical pathway PP2A-CDCA2/PP1-H3T3P-CPC-metaphase exit (major comment #2), and suggested them to look into the possibility that PBAF complex regulates spermatogenesis by change chromatin accessibility (major comment #3) giving the known function of this chromatin remodeling complex.

After reading the revised manuscript and the rebuttal letter, I felt that the authors did not do a good job in addressing these major issues. Although it is understandable that the rescue experiments (major comment #2) can be technically challenging and time consuming, the authors did not provide good explanations for the other two major comments either. In particular, their CUT&RUN interpretation is not convincing. Since CUT&RUN is one of their major evidences to support that Arid2 directly regulates genes known to govern spindle assembly/nuclear division, I am not convinced of their conclusion based on the CUT&RUN data provided. As shown below, although enrichment of both Arid2 and Brg1 are evident at Cdca2 promoter, their enrichment at other key genes including Aurka/Aurkb/Aurkc are not obvious (Aurkc shown below as an example). Thus, the major conclusion of this study "PBAF localizes to centromeres and promoters of genes known to govern spindle assembly and nuclear division" has no data to support.

For the minor comments, their co-IP data is still low quality and not convincing.

Reviewer #3 (Remarks to the Author):

In this revision the authors address many of my concerns and add a lot of compelling data. I still have some major concerns with the images and some suggestions.

Major concerns:

1. I still do not think that the IF with Arid2 and Brg1 in extended figure 1 is convincing. I agree that EF2 is convincing that Arid2 is absent. Spindle midzone staining usually appears as a sharp stripe, not diffuse cytosolic signal. Again, I strongly recommend removing this panel. The metaphase I arrest phenotype itself provides strong rationale for the study that the authors do not need to include weak localization data as part of their rationale. Ultimately it hurts their argument and is distracting.

2. Figure 5b- AURKA IF is not convincing in controls; I don't see an increase in abundance or on chromatin as stated- it looks like background. This IF data is not essential to their story, and I therefore recommend removing this since the expression of AURKA wasn't changed anyway. Ultimately it hurts their story line and is distracting. Removing it does not diminish the overall findings. If the authors are adamant to include AURKA analysis, I recommend trying the pAURKA antibodies that work better.

3. Figure 6- AURKC IF is also not convincing in controls; AURKB looks great and is convincing. The authors could use a pan pAURK antibody to look at changes of all 3 AURKs. It is possible that the AURKB antibody recognizes AURKC. If so, then the data could represent B/C. Maybe test specificity by western blotting. Is AURKB really diffusely spread and not just absent? Couldn't the AURKA-CPC claimed to be on chromatin be doing this? What about Borealin or survivin? Ultimately, this figure and interpretation needs to be tightened.

4. Extended figure 7b, row 4- The cdca2 pattern is described as diffuse, yet to me it looks more localized than in controls.

5. Figure 7a- I don't see chromatin localized cdca2 in control as described except in the 5%. Are they sure of the specificity? The pp1g data makes sense. Why not just show PP1g and eliminate Cdca2 for clarity?

Minor:

Description of 1b- include abbreviations in text for non-spermatocyte readership

Line 146- define P as phosphorylation

What is the difference b/t C2 and 3?

Author's response to points raised by reviewers

General Comments: We appreciate the opportunity to resubmit our manuscript. Major changes in the main text have been highlighted as requested. In this version, we have adequately addressed reviewer#2 and #3 concerns and provide further evidence that support our original conclusion that ARID2 containing PBAF complex facilitates spermatocyte division by coordinating the transcriptional activation of key cell division factors.

Briefly, improved CUT&RUN data analysis and visualization clearly show that several spindle assembly and cell division factors are bona fide PBAF targets. The new analysis includes peak calls using Sparse Enrichment Analysis for CUT&RUN (SEACR), a tool developed in Steve Henikoff's lab (Meers et al., 2019). Additionally, we reanalyzed previously published BRG1 ChIP-seq data generated from pachytene spermatocyte populations (Menon et al., 2019) to show significant enrichment at PBAF target genes identified by CUT&RUN. Therefore, the updated CUT&RUN analysis, coupled with the existing and new validation of candidate gene targets performed by monitoring cognate mRNA and protein abundances in response to the loss of ARID2 uphold our original conclusions. Furthermore, re-examination of the genomic data identified new PBAF target genes relevant to the spindle assembly defects seen in *Arid2^{ckO}* metaphase-I spermatocytes. Candidate targets include genes that encode, BORA, a known AURKA cofactor required for normal spindle assembly and CDC27, a core subunit of the Anaphase Promoting Complex (APC). These are very interesting candidates because BORA and APC are known to regulate AURKA activity and abundance, respectively. *Arid2^{ckO}* metaphase-I spermatocytes display reduced BORA and CDC27 abundance. These data fit nicely with our previous observations related to AURKA, which is mis-localized in the absence of ARID2. Furthermore, activated AURKA (pT288-AURKA), which is required for microtubule synthesis is moderately reduced and displaced from centrosomes, which provides greater insight into the mechanism underlying the role of ARID2 in spindle assembly.

While our manuscript has been under peer review, two new studies have shown that PBAF regulated genes, namely, *Plk1* and *Aurka* are required for centrosome formation and therefore spindle assembly in mouse spermatocytes (Alfaro et al., 2021; Wellard et al., 2021). These reports further support our model for a PBAF directed regulation of genes essential for metaphase progression in spermatocytes. Considering the recent evidence and our efforts to improve the quality and veracity of our conclusions, we believe that our manuscript merits a favorable decision.

Specific Responses to Reviewers Comments

Reviewer #1 (Remarks to the Author):

This reviewer thanks the authors for addressing my comments experimentally or by explanation/clarification. This is a nice story that expands the functional role of ARID2-containing PBAF complex in meiotic division. Congratulations. Rugang Zhang

We thank the reviewer for their kind words and appreciate the helpful comments that improved the manuscript.

Reviewer #2 (Remarks to the Author):

This is a revised manuscript. In the previous submission, I raised three major concerns regarding the potential mechanism leading to the observed phenotype (major comment #1), requested a rescue experiment to support their proposed hierarchical pathway PP2A-CDCA2/PP1-H3T3P-CPC-metaphase exit (major comment #2), and suggested them to look into the possibility that PBAF complex regulates spermatogenesis by change chromatin accessibility (major comment #3) giving the known function of this chromatin remodeling complex.

After reading the revised manuscript and the rebuttal letter, I felt that the authors did not do a good job in addressing these major issues. Although it is understandable that the rescue experiments (major comment #2) can be technically challenging and time consuming, the authors did not provide good explanations for the other two major comments either. In particular, their CUT&RUN interpretation is not convincing. Since CUT&RUN is one of their major evidences to support that Arid2 directly regulates genes known to govern spindle assembly/nuclear division, I am not convinced of their conclusion based on the CUT&RUN data provided. As shown below, although enrichment of both Arid2 and Brg1 are evident at Cdca2 promoter, their enrichment at other key genes including Aurka/Aurkb/Aurkc are not obvious (Aurkc shown below as an example). Thus, the major conclusion of this study “PBAF localizes to centromeres and promoters of genes known to govern spindle assembly and nuclear division” has no data to support.

For the minor comments, their co-IP data is still low quality and not convincing.

In the current revision, we have plotted the corresponding $\log_2(\text{CUT\&RUN}/\text{no IgG})$ values binned across the genome (Fig. 4b, Supplemental Fig. 6) C&R data presented in this fashion clearly identifies ARID2 and BRG1 enrichment at key cell division genes. We also call ARID2 peaks at cognate genes using SEACR, a tool developed for CUT&RUN data by the Henikoff group (Supplemental Fig. 6). As an additional measure of support, we re-examined BRG1 ChIP-seq data previously generate from P18 testes (normally enriched for pachytene spermatocytes) to show the presence of BRG1 at PBAF target genes (Supplemental Fig. 4c). Together these data validate PBAF associations with candidate genes.

We strongly disagree with several of the reviewer’s assessments:

1. “Since CUT&RUN is one of their major evidences to support that Arid2 directly regulates genes known to govern spindle assembly/nuclear division”.

In addition to CUT&RUN we thoroughly validated PBAF target genes relevant to the mutant phenotypes described in the manuscript. Validation was performed by monitoring the mRNA and protein abundance in response to the loss of ARID2 in metaphase-I spermatocytes. In addition to immunofluorescence, we provide western blotting data to support the mis-expression of candidate PBAF target genes (Supplementary Fig. 7b). We present new data to show that other target such as *Bora* (AURKA cofactor) and *Cdc27* (APC member) are mis-regulated in the absence of ARID2 (Fig. 5c, Supplementary Fig. 9). These factors are known to play a key role in regulating the activity and abundance of centrosomal member, AURKA, thereby providing further evidence of PBAFs role in spindle assembly in a transcriptional manner. Very interestingly, in support of our studies, two recent papers show that PBAF targets, *Plk1* and *Aurka* regulate centrosome formation in mouse spermatocytes (Alfaro et al., 2021; Wellard et al., 2021) (PMIDs: 33615678 & 33615693). Our data support the conclusions that PBAF governed gene regulation is essential for metaphase-I progression.

2. “Thus, the major conclusion of this study “PBAF localizes to centromeres and promoters of genes known to govern spindle assembly and nuclear division” has no data to support.”.

As stated in our response above the CUT&RUN identify PBAF associations with promoters of genes involved in cell division. As for his/her critique of PBAF centromeric association, we ask that the reviewer refer to Fig .4c, which unambiguously illustrates the enrichment of ARID2 and BRG1 at centromeric satellites.

3. “and suggested them to look into the possibility that PBAF complex regulates spermatogenesis by change chromatin accessibility (major comment #3) giving the known function of this chromatin remodeling complex”

Please refer Fig. 3c-d. Using MNase sensitivity as a measure of chromatin accessibility, we have demonstrated that chromatin is highly compacted in the absence of ARID2.

4. “For the minor comments, their co-IP data is still low quality and not convincing.”.

We are not sure why the reviewer remains unconvinced. The Co-IP blots clearly support interactions between PBAF (ARID2 & BRG1) (Fig. 2c) with PLK1 and PP2A (PPP2R1A & PPP2CA) (Fig. 8a). Additionally, the interaction between BRG1 and PPP2R1A is supported by our previous study where it was identified in spermatocytes by IP-Mass spectrometry, a technique with greater sensitivity than western blot (Menon et al., 2019).

Reviewer #3 (Remarks to the Author):

In this revision the authors address many of my concerns and add a lot of compelling data. I still have some major concerns with the images and some suggestions.

Major concerns:

1. I still do not think that the IF with Arid2 and Brg1 in extended figure 1 is convincing. I agree that EF2 is convincing that Arid2 is absent. Spindle midzone staining usually appears as a sharp

stripe, not diffuse cytosolic signal. Again, I strongly recommend removing this panel. The metaphase I arrest phenotype itself provides strong rationale for the study that the authors do not need to include weak localization data as part of their rationale. Ultimately it hurts their argument and is distracting.

We have removed this figure from the revised manuscript.

2. Figure 5b- AURKA IF is not convincing in controls; I don't see an increase in abundance or on chromatin as stated- it looks like background. This IF data is not essential to their story, and I therefore recommend removing this since the expression of AURKA wasn't changed anyway. Ultimately it hurts their story line and is distracting. Removing it does not diminish the overall findings. If the authors are adamant to include AURKA analysis, I recommend trying the pAURKA antibodies that work better.

The AURKA IF for controls is comparable to published literature (Wellard et al., 2020; Wellard et al., 2021) (see figures below; Journal of Cell Science (2020) 133, jcs248831. doi:10.1242/jcs.248831, EMBO reports 10.15252/embr.202051023).

The influence of ARID2 on AURKA is significant to this study for the following reasons:

1. Recent studies have highlighted the importance of AURKA in centrosome formation and spindle assembly in spermatocytes (Wellard et al., 2021) and oocytes (Blengini et al., 2021).
2. In the absence of ARID2 we observed a failure to form centrosomes and the concomitant mis-localization of AURKA. Quantification of AURKA fluorescence shows an increase in abundance in mutants relative to controls (Supplementary Fig. 8a, b). However, we find that activated AURKA (PT288-AURKA) is displaced from centrosomes and appears moderately reduced in the absence of ARID2.
3. Careful examination of our CUT&RUN data revealed that gene encoding BORA, a known regulator of AURKA localization and activity is a PBAF target. BORA levels are reduced in the absence of ARID2 (Fig. 5c, Supplementary Fig. 6-7). This deficiency in BORA potentially results in defects in AURKA localization and activity and therefore spindle assembly in the absence of ARID2. Overall, these data, add to the mechanism underlying the influence of ARID2 on spindle assembly.
4. Furthermore, ARID2 is also required for activation of PBAF target, *Cdc27* (*Apc3*), a core subunit of the APC (Anaphase Promoting Complex), known to target AURKA proteolysis

during metaphase (Supplementary Fig. 6-7, 9). Therefore, deficiencies in APC activity potentially results in the accumulation of AURKA.

3. Figure 6- AURKC IF is also not convincing in controls; AURKB looks great and is convincing. The authors could use a pan pAURK antibody to look at changes of all 3 AURKs. It is possible that the AURKB antibody recognizes AURKC. If so, then the data could represent B/C. Maybe test specificity by western blotting. Is AURKB really diffusely spread and not just absent? Couldn't the AURKA-CPC claimed to be on chromatin be doing this? What about Borealin or survivin? Ultimately, this figure and interpretation needs to be tightened.

1. AURKC localization was re-examined by IF using another antibody known to be specific to AURKC (Tang et al., 2006). The quality of the images generated with this antibody were superior to the previous images taken. The conclusion remained unchanged. We clearly show that chromatin bound AURKC is absent upon the loss of ARID2 (Fig. 6a), which is consistent with the repression of *Aurkc* in mutant spermatocytes (Supplementary Fig. 6-7).

2. The AURKB antibody used in this study has been shown not to recognize

AURKC [See figure below (column 3) from (Wellard et al., 2020)]. By western blotting AURKB levels are moderately reduced commensurate with its mRNA abundance in the absence of ARID2 (Supplementary Fig. 7a,b). The diffuse spreading might be reflective of slow AURKB turnover kinetics.

3. "What about Borealin or survivin?" The data clearly show that ARID2 activates AURKC and AURKB. The take home message is that ARID2 influences CPC function by regulating the expression of target genes, *Aurkc* and *Aurkb*. We therefore do not believe it is necessary to further investigate the localization of other CPC members, such as Borealin and Survivin.

4. Extended figure 7b, row 4- The *cdca2* pattern is described as diffuse, yet to me it looks more localized than in controls.

We have changed the figure label to reflect that CDCA2 is localized to chromatin

5. Figure 7a- I don't see chromatin localized *cdca2* in control as described except in the 5%. Are they sure of the specificity? The pp1g data makes sense. Why not just show PP1g and eliminate *Cdca2* for clarity?

There is moderate to minor CDCA2 overlap with chromatin in control and internal control, respectively. The chromatin localization is weak in most controls but not absent.

Reviewer1 required that we perform CDCA2 and H3T3P co-staining.

Minor:

Description of 1b- include abbreviations in text for non-spermatocyte readership

This has been changed

Line 146- define P as phosphorylation

This has been changed

What is the difference b/t C2 and 3?

Both C1-2 & 3 include regions with weak to no ARID2 occupancy based on k-means clustering of ARID2 enrichment. In the revised manuscript we have grouped these clusters together.

References:

- Alfaro, E., López-Jiménez, P., González-Martínez, J., Malumbres, M., Suja, J. A. and Gómez, R.** (2021). PLK1 regulates centrosome migration and spindle dynamics in male mouse meiosis. *EMBO Rep.*
- Blengini, C. S., Ibrahimian, P., Vaskovicova, M., Drutovic, D., Solc, P. and Schindler, K.** (2021). Aurora kinase A is essential for meiosis in mouse oocytes. *PLoS Genet.*
- Meers, M. P., Tenenbaum, D. and Henikoff, S.** (2019). Peak calling by Sparse Enrichment Analysis for CUT&RUN chromatin profiling. *Epigenetics and Chromatin.*
- Menon, D. U., Shibata, Y., Mu, W. and Magnuson, T.** (2019). Mammalian SWI/SNF collaborates with a polycomb-associated protein to regulate male germline transcription in the mouse. *Development* **146**, dev174094.
- Tang, C. J. C., Lin, C. Y. and Tang, T. K.** (2006). Dynamic localization and functional implications of Aurora-C kinase during male mouse meiosis. *Dev. Biol.*
- Wellard, S. R., Schindler, K. and Jordan, P. W.** (2020). Aurora B and C kinases regulate chromosome desynapsis and segregation during mouse and human spermatogenesis. *J. Cell Sci.* **133**, jcs248831.
- Wellard, S. R., Zhang, Y., Shults, C., Zhao, X., McKay, M., Murray, S. A. and Jordan, P. W.** (2021). Overlapping roles for PLK1 and Aurora A during meiotic centrosome biogenesis in mouse spermatocytes. *EMBO Rep.*

REVIEWERS' COMMENTS

Reviewer #2 (Remarks to the Author):

This version of the manuscript has improved. However, the authors did not address my concern on the CUT&RUN analyses. The major evidence that the authors used to support the direct involvement of Arid2 in regulating spindle assembly/nuclear division is their claim that Arid2 directly binds to the genes involved in spindle assembly/nuclear division based on their CUT&RUN data. However, the CUT&RUN data presented does not support this claim as I pointed out in my last round of review. The authors still only show a very narrow window, make it hard to evaluate the peak quality. Indeed, it is surprising that Aurkb has comparable Arid2 enrichment signals in both P28 and P28.5 tracks (Fig. S6), but only the P28 sample was labeled to have an Arid peak (solid black bar). Similar concern also apply to gene Ppp2ca.

While I believe that Arid2 does regulate some genes important for spindle assembly/nuclear division, whether this regulation is direct as the authors claimed remains to be shown with convincing CUT&RUN data. In my opinion, only Cdca2 has convincing Arid2 binding as revealed by the tracks I pointed out in my last round of review, other targets, like AURKC and CDC27, are not convincing, which is consistent with the fact that their protein levels are not changed in Western Blotting results (sFig7B).

Reviewer #3 (Remarks to the Author):

Thank you for your thoughtful modifications and responses. I'm confused by response 2 regarding Figure 5b. The IF you are responding to is for AURKA, but your justification is showing AURKB IF from the Wellard paper. Your modifications in the figure change to pAURKA but that is not detailed here. Perhaps this was the wrong response version? Regardless, I am pleased with the modifications.

Response to reviewer comments:
REVIEWERS' COMMENTS

Reviewer #2 (Remarks to the Author):

This version of the manuscript has improved. However, the authors did not address my concern on the CUT&RUN analyses. The major evidence that the authors used to support the direct involvement of Arid2 in regulating spindle assembly/nuclear division is their claim that Arid2 directly binds to the genes involved in spindle assembly/nuclear division based on their CUT&RUN data. However, the CUT&RUN data presented does not support this claim as I pointed out in my last round of review. The authors still only show a very narrow window, make it hard to evaluate the peak quality. Indeed, it is surprising that *Aurkb* has comparable Arid2 enrichment signals in both P28 and P28.5 tracks (Fig. S6), but only the P28 sample was labeled to have an Arid peak (solid black bar). Similar concern also apply to gene *Ppp2ca*.

While I believe that Arid2 does regulate some genes important for spindle assembly/nuclear division, whether this regulation is direct as the authors claimed remains to be shown with convincing CUT&RUN data. In my opinion, only *Cdca2* has convincing Arid2 binding as revealed by the tracks I pointed out in my last round of review, other targets, like *AURKC* and *CDC27*, are not convincing, which is consistent with the fact that their protein levels are not changed in Western Blotting results (sFig7B).

Response:

In this study we provide sufficient evidence to show that ARID2 influences the expression of spindle assembly/nuclear division genes, as the reviewer has stated. With respect to its direct involvement in gene regulation, we identified ARID2 enrichment at transcription start sites (TSSs) of genes involved in cell division. Crucial genes identified in this manner also display peaks identified by SEACR. We agree that future validation using a CUT&RUN control such as *Arid2^{cKO}* will be important in

validating ARID2 genomic associations and is acknowledged in the revised manuscript. In fact, from preliminary ARID2 CUT&RUN data where we have monitored its signal in *Arid2^{WT}* and *Arid2^{cKO}* spermatocytes isolated from P19-P20 testes (Figure panel on right), candidate targets such as *Aurkb* display decreased ARID2 signal in mutant relative to control sample. This difference is less stark in the case of *Ppp2ca*, which would be consistent with a lack of its mis-expression upon the loss of ARID2 (Supplementary Fig.6). Since these studies are

preliminary in nature and lack replicates, we have not included it in the manuscript but instead present these data in our response to allay the reviewer's concerns.

Additionally, we have the following comments on concerns raised by the reviewer:

1. Fewer ARID2 peaks were called with P28.5 samples relative to P28. This also agrees with fewer TSS's displaying enrichment of ARID2 and BRG1 at P28.5 relative to P28 (Figure. 4). SEACR peaks are based on setting a threshold at which the percentage of target regions/blocks versus control (no IgG) regions/blocks is maximal. Only target regions/blocks that exceed this threshold are called peaks. Differences in thresholding at P28 and P28.5 contribute to the variability in peak calling at distinct stages. In the case of *Aurkb*, ARID2 enrichment underlying the peak is lesser at P28.5 relative to P28 (Supplementary Fig.6). For *Ppp2ca*, it is possible that the apparent peakiness at P28.5 is a false positive. In fact, in the manuscript we conclude that the reduction of PPP2CA levels in response to the loss of ARID2 appears not to occur from transcriptional mis-regulation but rather a consequence of altered protein stability. SEACR has been shown to be better at avoiding false positives relative to other peak callers. Off course, this will need to be confirmed using *Arid2* knockout samples as indicated by our preliminary data (shown above). Nonetheless all candidate genes that were investigated in this study display ARID2 enrichment at their TSSs and promoter peaks at least at P28. Also, expression of candidate genes in response to the loss of ARID2 were thoroughly validated by other methods described in the paper and the results were consistent a metaphase-I arrest phenotype.
2. With respect to the reviewer's comment about showing only a narrow window of the browser shots in supplementary fig.6, we have shown a heatmap displaying CUT&RUN enrichment over a 4Kb window centered at TSSs. The browser view in supplementary fig.6 displays a 2 Kb window around the promoter, allowing us to accommodate panels for important candidate genes. Furthermore, access to the genome browser tracks were made available in the reporting summary.
3. With respect to the reviewer's comments about CDC27 and AURKC, we clearly show a loss of protein abundance in mutant metaphase-I spermatocytes relative to controls (Supplementary figure. 9a, Figure. 6a). Furthermore, by quantification of western blots from control and mutant testes (from n=3), the average abundance of AURKC level is reduced (Supplementary Fig 7b, left panel, source data).

Reviewer #3 (Remarks to the Author):

Thank you for your thoughtful modifications and responses. I'm confused by response 2 regarding Figure 5b. The IF you are responding to is for AURKA, but your justification is showing AURKB IF from the Wellard paper. Your modifications in the figure change to pAURKA but that is not detailed here. Perhaps this was the wrong response version? Regardless, I am pleased with the modifications.

We thank the reviewer for constructive comments that have helped improve the quality of the paper. We apologize for forgetting to mention the new pAURKA IF data in Figure. 5b that was included in the revised manuscript. The AURKA data was moved to supplementary fig. 8 in the revised manuscript.

I think the reviewer is confusing two separate points in our previous response.

In point #2, we included a panel from the Wellard et al. paper to illustrate that the AURKA localization we observed in metaphase-I spermatocytes was similar to their published data.

In point #3, I cited data from another study by Wellard and colleagues, showing the knockout validation of the AURKB antibody we used in this study.